# Atmospheric bromoform at Cape Point, South Africa: An initial fixed point dataset on the African continent

Brett Kuyper[1,*], Carl J. Palmer[1,2], Casper Labuschagne[3], and Chris J. C. Reason[1]

[1]Department of Oceanography, University of Cape Town, Cape Town
[2]Applied Centre for Climate and Earth System Science, CSIR, Rosebank
[3]South African Weather Service, Stellenbosch
[*]Now at: Department of Chemistry, University of the Western Cape, Cape Town

*Correspondence to:* Brett Kuyper (3479857@myuwc.ac.za)

**Abstract.** Bromoform mixing ratios in marine air were measured at Cape Point Global Atmospheric Watch Station, South Africa. This represents the first such bromoform data set recorded at this location. Manual daily measurements were made during a month long field campaign (austral spring 2011) using a gas chromatograph-electron capture detector (GC-ECD) with a custom built front end thermal desorption trap. The measured concentrations ranged between 4.4 and 64.6 ($\pm$ 22.2 %) ppt with a mean of 24.8 $\pm$ 14.8 ppt. The highest mixing ratios recorded here occurred at, or shortly after, low tide. The diurnal cycle exhibited a morning and evening maximum with lower concentrations throughout the rest of the day. Initial analysis of the data presented indicate that the local kelp beds were the dominant source of the bromoform reported. A concentration weighted trajectory analysis of the bromoform measurements suggests that two offshore source areas may exist. These source areas appear to be centred on the Agulhas retroflection and extending from St Helena Bay to the southwest.

## 1 Introduction

### 1.1 Bromoform in the marine environment

Bromoform ($CHBr_3$) is a brominated methane-like hydrocarbon which is a volatile liquid at room temperature. Bromoform is naturally produced by kelp and phytoplankton in the upper layers of the ocean (Quack and Wallace, 2003). A few anthropogenic sources are known including water treatment, nuclear power plants (Quack and Suess, 1999) and rice paddies (Redeker et al., 2003), however these tend to small on a global scale (Carpenter and Liss, 2000; Quack and Wallace, 2003). It was estimated that globally between 2.2 x $10^{11}$ - 2.5 x $10^{12}$ g $CHBr_3$ yr$^{-1}$ (Carpenter and Liss, 2000; Quack and Wallace, 2003) is produced of which only 3.0 x $10^{10}$ g $CHBr_3$ yr$^{-1}$ is anthropogenic (Gschwend et al., 1985; Allonier et al., 1999; Quack and Wallace, 2003), the rest being from natural sources, including 1.3 x $10^{11}$ g $CHBr_3$ yr$^{-1}$ from brown algae and 1.7 - 2.0 x $10^{11}$ g $CHBr_3$ yr$^{-1}$ from phytoplankton (Manley et al., 1992; Carpenter and Liss, 2000; Quack and Wallace, 2003). Outgassing to

the atmosphere constitutes the largest known oceanic loss of bromoform, which is relatively stable to chemical loss pathways (hydrolysis and nucleophilic substitution) in seawater at ambient temperatures (Carpenter and Liss, 2000; Quack and Wallace, 2003; Jones and Carpenter, 2005). The production of bromoform in the oceans forms an important step in the biogeochemical cycling of bromine through the Earth system (Warwick et al., 2006; Hossaini et al., 2010).

The production of bromoform by phytoplankton and kelp has been shown to be stimulated through oxidative stress (Quack and Wallace, 2003; Palmer et al., 2005; Kupper et al., 2008) and a maximum rate has been linked with the photosynthetic cycle (Collén et al., 1994). However, the specific reasons for bromoform production in these organisms remains unknown (Moore et al., 1996; Paul and Pohnert, 2011; Kuyper, 2014). Production by kelp is thought to be the dominant natural bromoform source to the marine environment (Carpenter and Liss, 2000). Different species of kelp are known to produce bromoform at

varying rates (e.g. Nightingale et al., 1995). Laboratory studies have measured significantly higher mixing ratios from kelp, per weight, when compared to phytoplankton (Tokarczyk and Moore, 1994; Moore et al., 1996; Carpenter and Liss, 2000). However, kelp species are coastally constrained, while phytoplankton are able to cover hundreds of square kilometres (Jennings et al., 2001; Kudela et al., 2005). A question remains regarding the dominant contribution to the global bromoform budget.

## 1.2   Implications for atmospheric chemistry

The rate of outgassing to the atmosphere, gas flux rate, is proportional to the wind speed and the solubility of the gas (Liss and Merlivat, 1986; Wanninkhof, 1992; Nightingale et al., 2000). The majority of the outgassed bromoform remains below the tropopause, with a small amount escaping to the stratosphere (Warwick et al., 2006; Hossaini et al., 2010; Saiz-Lopez et al., 2012). Photolysis of bromoform is the dominant sink once in the atmosphere, which results in an atmospheric lifetime of 2-3 weeks (Carpenter and Liss, 2000; Quack and Wallace, 2003). The photolysis of bromoform releases bromine radicals

into the atmosphere. These bromine radicals are an important catalyst in the destruction of ozone in the upper troposphere and lower stratospheric region (Warwick et al., 2006; Hossaini et al., 2010). Ozone in this region plays two key functions: in the upper troposphere (UT) ozone is a potent greenhouse gas, whereas in the lower stratosphere (LS) it forms part of the ozone layer, absorbing incoming UV radiation (Saiz-Lopez et al., 2012). In the UT bromine radicals, released predominantly from bromoform, are known to catalytically react with ozone. This results in the destruction of the ozone and subsequent loss

from the region (Aschmann et al., 2009; Hossaini et al., 2010; Saiz-Lopez et al., 2012). Thus, bromine chemistry could play a significant role in climate change through ozone depletion in the UT (Hossaini et al., 2010; Saiz-Lopez et al., 2012).

Estimates have been made of both the amount of bromoform reaching the upper troposphere and the magnitude of the impact this has on climate change. These estimates are based on poorly constrained source emissions from the global ocean (Warwick et al., 2006; Hossaini et al., 2010). It is estimated that between 1.6 and 3.0 ppt of inorganic bromine is contributed directly

from bromoform to the lower stratosphere (Aschmann et al., 2009). The background atmospheric bromoform mixing ratios are estimated to be 1-2 ppt. However, local mixing ratios can be elevated above this. This typically occurs in regions with extensive kelp beds and in areas of strong coastal upwelling (e.g. Quack and Wallace, 2003; Quack et al., 2007a). The skill of atmospheric chemistry models would be greatly enhanced if there was better quantification of the source strength of bromoform, and in turn,

its impact on bromine radicals and ozone chemistry in different regions. Such enhancement of modelling capacity would lead to a vastly improved understanding of the roles of the source and product gases in the UT/LS region.

Quantifying the inventories of bromoform emissions is thus critical in better characterising the oxidative capacity of the atmosphere. This is particularly pertinent in the tropics, where deep convection results in a greater percentage of bromine radicals reaching the UT/LS region (Hossaini et al., 2010; Saiz-Lopez et al., 2012). Understanding the sources in the tropics is therefore of great specific scientific interest (Palmer and Reason, 2009). However, there exists a paucity of measurements of bromoform in the tropics (Palmer and Reason, 2009). Existing data in this region tend to be from transient ship cruises, which only provide a discrete snapshot at the point in space/time that the cruise transects the area of interest. Similarly, no time series of measurements at a fixed point currently exists for a coastal site in southern Africa. The Cape Point GAW monitoring station provides a point from which to begin addressing this lack of southern African measurements. Furthermore, the Cape Point monitoring station fills a critical Southern Hemisphere latitudinal gap between Cape Matatula, American Samoa (14 °S) and Cape Grim, Tasmania (41 °S) (Brunke and Halliday, 1983).

## 1.3 Significance of Cape Point location

Here we present the first ever bromoform dataset recorded at the Cape Point Global Atmospheric Watch (GAW) station (34.3 °S 18.5 °E, Fig. 1). This station offers a unique location from which to measure bromoform mixing ratios in a subtropical region, but is also suitable to sample air from the south Atlantic and Southern Ocean. Wind direction and radon concentration ($^{222}$Rn) at Cape Point have been extensively used to classify the arriving air masses (Brunke et al., 2004; Whittlestone et al., 2009). A mixture of air sources have been recorded at Cape Point. Brunke et al. (2004) classify these as follows: 100 % clean marine (baseline, $^{222}$Rn $< 350\,\mathrm{mBq\,m^{-3}}$) to 100 % continental (with/without anthropogenic influence, $^{222}$Rn $>1500\,\mathrm{mBq\,m^{-3}}$) and intermediate (mixture of baseline and continental, $800 < {}^{222}$Rn $< 1500\,\mathrm{mBq\,m^{-3}}$). The subtropical location of Cape Point may make this region a particularly significant source of bromoform to the atmosphere, specifically when considering the potential impact on global ozone budgets. The region lies in close proximity to the tropics where deep convection is able to rapidly transport the outgassed bromoform into the UT/LS, where bromine initiated catalytic ozone destruction occurs. To quantify the importance of the measurements made at Cape Point to tropical deep convection it is necessary to note how the synoptic conditions change seasonally over South Africa. During summer approximately 5 % of trajectories from South Africa escape to the Atlantic (10 °S), while 75 % of transport exits to the southeast (Tyson and Preston-Whyte, 2000). Ridging high pressure systems, present during spring and autumn, increase the transport to the tropical Atlantic to 25 % (Tyson and Preston-Whyte, 2000). Moreover, data recorded here is of particular value as the size of the contribution from the Cape Point region is to date largely untested. The Cape Point data presented here represent the first of their kind in Africa, or for the south Atlantic region (Cox et al., 2003).

The Southern Ocean is largely regarded as a highly biologically active region, especially during the spring and summer (Arrigo et al., 2012). This region may provide a significant contribution to the global atmospheric loading of bromoform. However, the Southern Ocean is widely under-sampled when it comes to bromoform measurements. Although there have been sporadic ship cruises to the Southern Ocean (Ziska et al., 2013), no long term work has been done in the Atlantic sector of

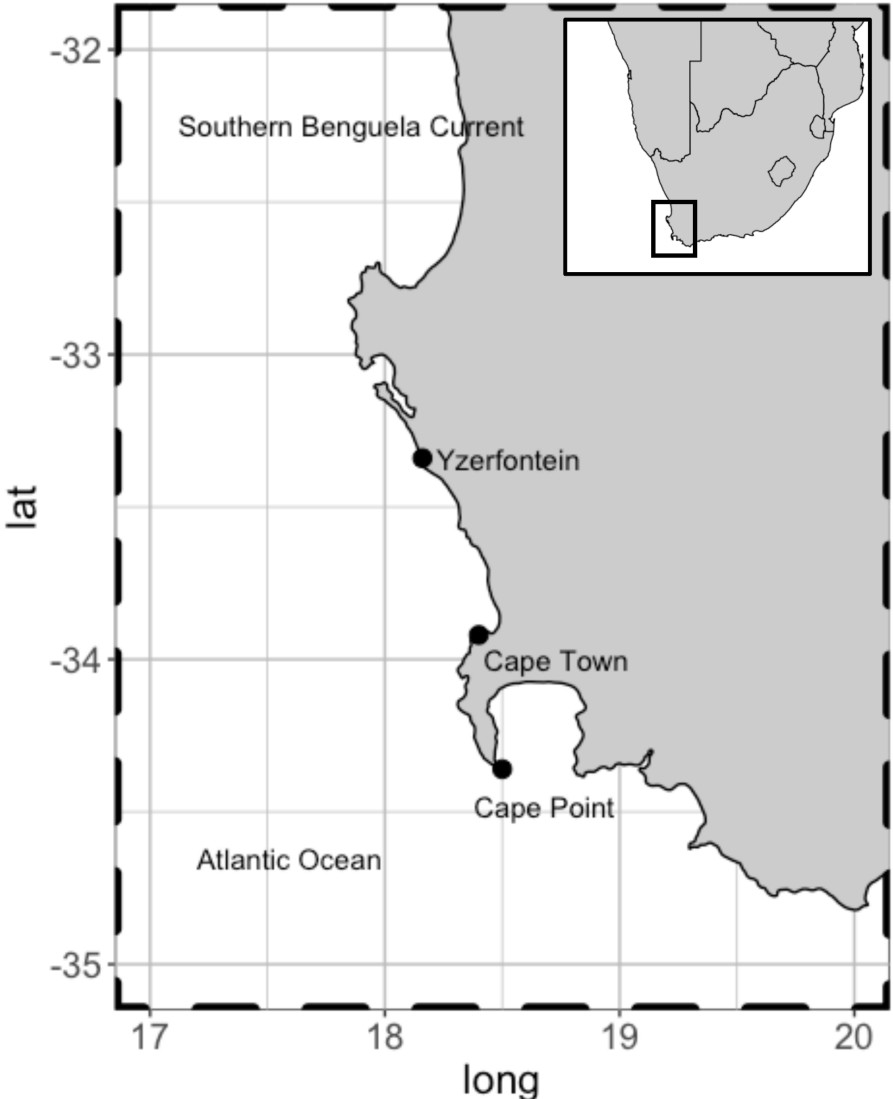

**Figure 1.** Location of Cape Point in relation to Cape Town. Kelp range along the entire coast. These are dominated by *Ecklonia maxima Papenfuss* south of Yzerfontein but transition to predominantly *Laminaria pallid Greville* north of Yzerfontein. Adapted from Kuyper 2014

the Southern Ocean. The data presented here therefore offer the first fixed point measurements of bromoform in air from the Atlantic sector of the Southern Ocean.

In addition to receiving baseline air from the south Atlantic and Southern Ocean, Cape Point lies in close proximity to extensive kelp beds. The kelp beds extend along the South African coast to the north and east of Cape Point. A variety of remote sensing techniques have been used to assess the extent and composition of kelp beds in 19 predefined areas along the Cape coast (Anderson et al., 2007). The studies have shown that kelp beds are present in all 19 areas ranging from a minimum of 11

ha coverage in Table Bay to a maximum of just under 1000 ha north towards the Namibia border. The species composition was predominantly *Ecklonia maxima Papenfuss* south of Yzerfontein, but transitioned to predominantly *Laminaria pallid Greville* north of Yzerfontein (Fig. 1). Thus, Cape Point is an ideal location to sample the open ocean, local tidally affected kelp beds, as well as the occasional anthropogenic pollution event from the greater Cape Town region; based on the seasonally varying wind direction. Addressing the paucity of data from this region will be instrumental in separating the persistent conundrum as to the major source of bromoform in the atmosphere.

## 2 Methods

The separation, identification and quantification of bromoform was achieved using a gas chromatograph (GC) with an electron capture detector (ECD) system. This featured a custom built thermal adsorption/desorption trap for the pre-concentration of atmospheric samples and delivery of analytes onto the GC column (Kuyper et al., 2012; Kuyper, 2014). Specific details of the sampling method in this campaign are described below.

### 2.1 Sampling

The measurements of bromoform were made at the Cape Point Global Atmospheric Watch station in the austral spring of October and November 2011. The GAW station sits at the top of a coastal cliff (230 m a.s.l) at the end of a peninsula south of Cape Town (Fig. 1). The manual nature of the GC system, coupled with periods of instrument downtime, resulted in a quasi-continuous sampling pattern with a measurement frequency of approximately 45 $min$ to 1 $hour$. A total of 135 discrete bromoform measurements were made in air samples during this period.

A Shimadzu GC-8A with a Perkin Elmer F-22 ECD was used to record the bromoform concentrations. A J & W Scientific DB-624 (30 m x 320 x 1.8 μm, 5 % polarity film) capillary column was used in the oven to achieve the separation of samples (Itoh et al., 1997). A 30 $ml\,min^{-1}$ nitrogen flow was added directly to the ECD in the form of make up gas. Helium (Grade 5.0, Air Liquide) at a constant flow rate of 5 $ml\,min^{-1}$ was maintained maintained through the column at the start of the each analysis. The oven was held at 35 $°C$ for 5 $min$ following the injection of a sample. Thereafter, the temperature was increased to 60, 90, 150, and 200 $°C$ every 5 $min$. The temperature in the oven was increased at 65 $°C\,min^{-1}$ and held isothermally once the new temperature was reached.

Air samples were pre-concentrated in a custom built thermal desorption unit (TDU, Kuyper et al., 2012). Adsorbents (Carbopac X and Carboxen 1016, 9 mg each) held in a glass tube were cooled to -20 $°C$ during the trapping phase. The cooling of the system was achieved by a recirculating chiller filled with glycol. To exclude air from the adsorbent trap a flow of helium (100 $ml\,min^{-1}$, Grade 5.0) was maintained both before and after sampling. Samples were dried using magnesium perchlorate, held in a glass moisture trap, before being passed to the trap, as per Groszko and Moore (1998). Air was passed through the adsorbent trap at 100 $ml\,min^{-1}$ for 15 $min$, resulting in a 1.5 l sample volume. The sampling flow rate was checked weekly by means of a digital flow meter. An oil free piston pump was used to draw air through a 60 $m$ Decabon sampling line and the adsorbent trap. This was routed through a T-piece with the excess gas vented to the atmosphere. A mass flow controller was

used to regulate the gas flow through the adsorbent trap. The pump was operated at $400 \, \mathrm{ml \, min^{-1}}$ and a needle valve on the exhaust was used to ensure sufficient pressure in the sampling line for the mass flow controller to operate.

A built in resistance wire heated the TDU glass tube to $400 \, °\mathrm{C}$ to desorb samples for injection. A second stage cryo-focusing system was used at the head of the column, with liquid nitrogen, to improve the chromatography. The liquid nitrogen was held at the head of the column for the duration of the primary injection. Thereafter, boiled water was used to desorb the samples trapped at the head of the column.

## 2.2 Calibration

An external calibration method was used to verify the system performance. A custom built permeation oven was used to deliver aliquots of bromoform at varying concentrations to the trap (Wevill and Carpenter, 2004; Kuyper, 2014). A bromoform permeation tube held at $70 \, °\mathrm{C}$ (permeating at $343 \, \mathrm{ng \, min^{-1}}$) was flushed with nitrogen (grade 5.0, Air Liquide) at $100$ $\mathrm{ml \, min^{-1}}$. This gas mixture was continually passed through a $100 \, \mu l$ sample loop and exhausted through a halocarbon trap. Aliquots of 100-300 μl (1-3 sample loops) of the resulting permeation gas (bromoform diluted in nitrogen), were introduced to the thermal desorption unit from the permeation oven. Calibration samples were passed through the drying trap as for air samples, thus any loss would be consistent for air and calibration methods. The calibration points were analysed using the same temperature programme as air samples to ensure identical retention times. These were also used for the identification of bromoform.

A complete calibration curve (Fig. 2) was measured prior to the start of the experimental period. The peak area was determined from the repeated injection of 1-3 loops of diluted bromoform in nitrogen gas. Peak areas were calculated through the trapezoid integration method and were computed in MATLAB (Poole, 2003). The mixing ratios of the injected loops were calculated from the number of moles of bromoform injected, as follows. Each loop injection resulted in 0.343 ng of bromoform being loaded on the trap, based on the calibrated rate of the permeation tube (Wevill and Carpenter, 2004; Kuyper, 2014). The number of moles of bromoform on the trap was calculated from this mass which resulted in $1.36 \, \mathrm{x10^{-12}}$ mol being loaded on the trap, per sample loop injection. The number of molecules of bromoform was calculated by multiplying the number of moles by the Avogadro constant to yield the number of bromoform molecules on the trap. The total number of molecules in a sample was calculated by multiplying the air number density ($2.5 \, \mathrm{x10^{25} \, molecules \, m^{-3}}$) with the sample volume (1.5 l). The bromoform mixing ratio of one loop was calculated as the number of bromoform molecules of one loop as a fraction of number of molecules in a sample multiplied by $10^{12}$ to yield ppt.

A complete system calibration was performed at the start of the sampling at Cape Point (12/13 October 2011). Thereafter, a calibration point of 1-3 loops was measured approximately daily to account for system drift. After the initial calibration the daily calibration points were coerced into a regular matrix, with 8 hour time steps. This resulted in three calibration points per day. Gaps between calibration points were interpolated using a three point running mean. An overall $\mathrm{r^2}$ of 0.82 between the peak area and mixing ratio was achieved using this system during the sampling period (Fig. 2). An analysis of peak area of repeated two loop injections indicated a system precision of 22.2 % based on the RSD, including the running mean estimates. Following an analysis of the calibration curve a limit of detection of 0.21 ppt was determined for this system. An interquartile

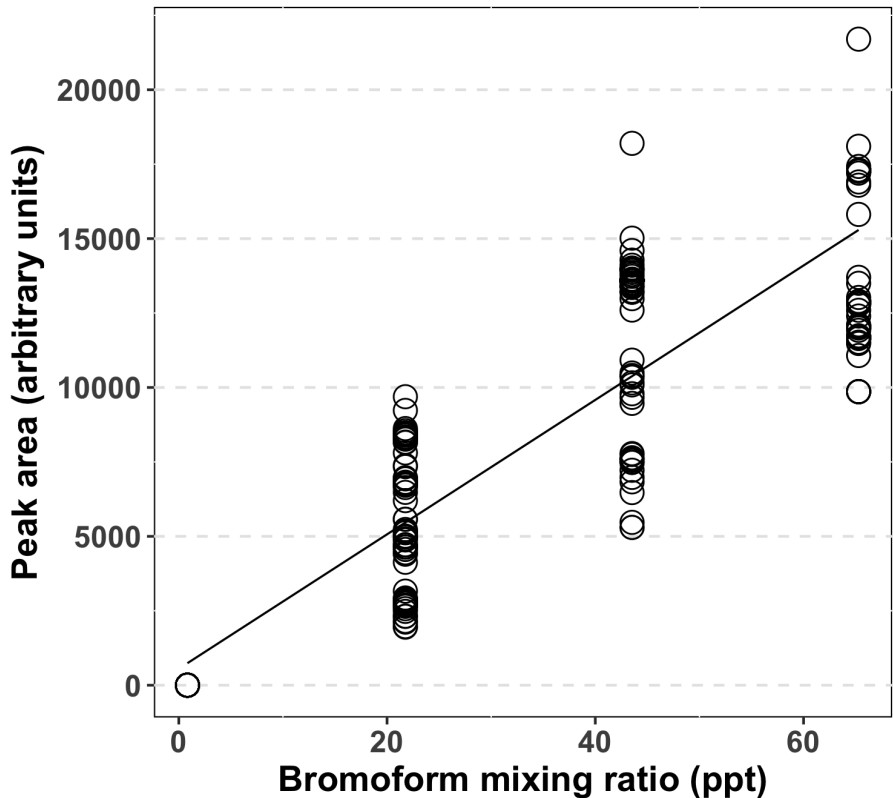

**Figure 2.** Combined calibration data for the GC-ECD system for bromoform, based on multiple loop injections and interpolation over the entire sampling period. Peak area calculated using the trapezoid method.

range (IQR) method was used to search for and remove outlying data within the data set (Underhill and Bradfield, 2005). The IQR is the difference between the 25 % and 75 % quartiles. This is then multiplied by 1.5 and either added to or subtracted from the 75 % or 25 % quartile, respectively. Any value greater than the upper bound was removed, whereas the lower bound was below the LOD. The upper bound was calculated to be 71.4 ppt.

## 2.3 Ancillary measurements: Cape Point, Global Atmospheric Watch

The Global Atmospheric Watch (GAW) station at Cape Point is operated by the South African Weather Service. In addition to the standard meteorological parameters, numerous climate relevant gases are quantitatively measured here, including: $CO_2$, $CH_4$, CO, radon ($^{222}Rn$) and $O_3$ (Whittlestone and Zahorowski, 1998; Brunke et al., 2004; Whittlestone et al., 2009).

Air samples were drawn in at the top of a 30 m high sampling mast. A continuous flow system was used in the laboratory to exclude the accumulation of any contamination. Sequential cold trapping at -5 and -40 °C along the flow path was used to dry air samples prior to measurement. A 30 min mean was applied to all data to standardise different sampling periods.

The ozone measurements were made on a Thermo Electron 49C analyser. These analysers are based on the UV absorption technique and calibrated every two months. Daily zero and span measurements were used to assess long-term stability of the detectors. A Trace Analytical RGA3 was used to measure atmospheric CO mixing ratios. The detector uses a reduction of mercuric oxide (HgO) to determine the concentration of CO (Brunke et al., 2004). A measurement was made every 15 $\min$

with a calibration occurring every 2 hours. Radon ($^{222}$Rn) measurements were made in an ANSTO-build, two-stage $\alpha$-decay system which detects the collected radon daughter products (Whittlestone and Zahorowski, 1998; Whittlestone et al., 2009). A sample was measured half-hourly and calibrated monthly.

## 2.4 Ancillary measurements: NOAA HYSPLIT model, Marine boundary layer height and Diurnal cycle

**NOAA Hysplit model**

The HYbrid Single-Particle Lagrangian Integrated Trajectory (HYSPLIT: http://www.ready.noaa.gov/) model was used in addition to the chemical tracers to examine the source of air masses being sampled (Stein et al., 2015). These trajectories were generated using the NCEP reanalysis data as the meteorological data in *R*. The back trajectories were calculated for 96 hours prior to bromoform measurement. The trajectories were merged with the bromoform data set to allow for integrated analysis using the *openair* package (Carslaw and Ropkins, 2012). This package comprises a range of statistical tools to examine back

trajectories, in order to identify source regions or contributions.

The potential source contribution function (PSCF) in *R openair* calculates the probability that a source exists at a specific location (Fleming et al., 2012; Pekney et al., 2006). The PSCF is calculated by the ratio of trajectories with elevated concentrations to the number of times those trajectories pass through a specific point, defined as grid cells. A value for each grid cell is calculated. These grid cell values can be the same when the sample concentrations are either marginally above or greatly

elevated from a defined criterion, e.g. the mean (Hsu et al., 2003; Carslaw and Ropkins, 2012). Consequently, the difference between strong and moderate sources can be difficult to distinguish. The concentration weighted trajectory (CWT) method can be used to potentially identify source areas through the calculation of concentration fields. The mean of the concentration for each grid cell was calculated as follows:

$$ln(\bar{C}_{ij}) = \frac{1}{\sum\limits_{k=1}^{N} \tau_{ijk}} \sum\limits_{k=1}^{N} ln(c_k)\tau_{ijk} \tag{1}$$

where *i* and *j* are the grid indices, *k* is the index of the trajectory, *N* is the total number of trajectories used, $c_k$ is the pollution concentration of trajectory *k* upon arrival and $\tau_{ijk}$ is the residence time of trajectory *k* in grid (*i, j*) (Carslaw and Ropkins, 2012). High concentrations at the measurement site would, on averaged, be caused by grid cells with elevated values of $\bar{C}_{ij}$. Thus, indicating possible source regions. The CWT back trajectory calculation was performed on the entire data set of bromoform measurements at Cape Point. As a first approximation of the offshore sources of bromoform to Cape Point a concentration

weighted trajectory (CWT) model analysis of the back trajectories was performed.

**Marine boundary layer (MBL) height**

Twice daily radiosondes were released from Cape Town international airport at local midnight and noon. The airport lies approximately 60 km northeast of Cape Point. The height of the MBL was determined by the surface and elevated temperature inversion methods from the radiosonde data (Seibert et al., 2000; Seidel et al., 2010). The calculated boundary layer height at the airport was used as a proxy for the marine boundary layer at Cape Point.

**Tidal height**

The tidal height for Cape Town was obtained from the South African Hydrographic Office (SAHO). Tidal gauges are used to measure the height in the harbours around South Africa. Due to periodic instrument failures of some of the gauges around Cape Town during the bromoform sampling period, tidal height estimates were used to interpolate over any gaps. The height is given in metres above a SAHO locally defined chart datum and therefore the lower the value the lower the tide.

**Diurnal cycle**

A mean diurnal cycle was calculated from the full range of Cape Point measurements using the *timeVariation* function of *openair* (Carslaw and Ropkins, 2012). The data was sorted into 24 hourly bins. The time of the sampling was used to assign an hourly bin to each measurement. The mean and 95 % confidence interval of the mean in each bin were then calculated.

## 3   Results and Discussion

The bromoform mixing ratios at Cape Point were measured in the range 4.4-64.6 ppt with a mean of $24.8 \pm 14.8$ ppt (Fig. 3). Bromoform was typically in the range of 4-20 ppt but on several occasions elevated mixing ratios were encountered that could last for several hours (Fig. 3). The range of variability observed at Cape Point is comparable to previously published work, specifically with reference to coastal sites (Table 1).

The measurements were made in a variety of air masses ranging from clean marine to continental air. This suggests that a number of sources may have impacted on the bromoform mixing ratios at Cape Point. Nearly 57 % of bromoform measurements recorded here, were below the mean. This indicates that the mean value is skewed by a few elevated bromoform mixing ratios. When examined over the whole data set the bromoform mixing ratios showed only weak correlations with the measured meteorological and physical measurements ($r^2 < 0.4$).

**Link to tidal cycle**

The full tidal spectrum was captured at Cape Point during the bromoform sampling period, including two neap tides and a spring tide. A maximum tidal range of approximately 2 m was observed during the spring tide. This range decreased to a maximum of 1 m during the neap tides (Fig. 3). Exposure of kelp (which as discussed is present in abundance at Cape Point) to the atmosphere at low tide has been linked with an increase in atmospheric bromoform mixing ratios, for example, a site at

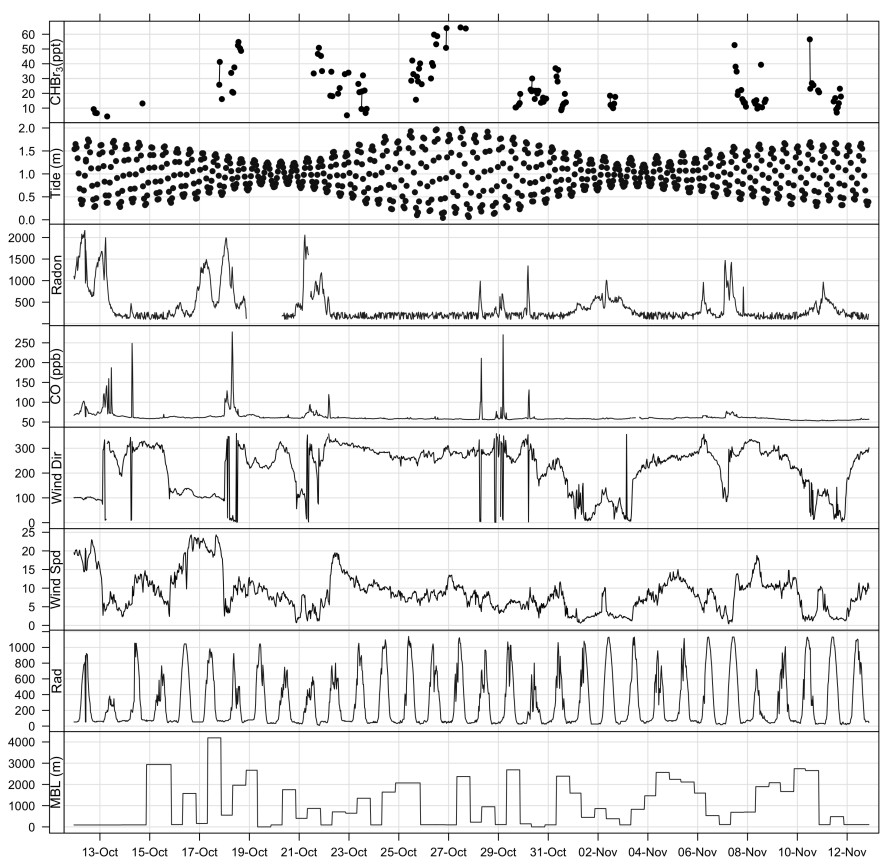

**Figure 3.** Time series plot of bromoform and meteorological measurements at Cape Point during October/November 2011. Tide height given in m above SAHO datum, Radon ($\mathrm{mBqm^{-3}}$), Wind direction in degrees, Wind speed ($\mathrm{m\,s^{-1}}$), Radiation ($\mathrm{Wm^{-2}}$).

which this has been observed is Mace Head on the west coast of Ireland (Carpenter et al., 1999). An increase in the oxidative stress on the kelp initiated by solar radiation is thought to drive this correlation (Carpenter et al., 1999; Palmer et al., 2005). However, the measured bromoform mixing ratios reported here do not correlate well with the tidal pattern. While the maximum tidal range in the vertical at Cape Point is comparable to that at Mace Head, the horizontal extent is much smaller. This may explain the lack of local correlation. Consequently, during low tide at Cape Point, only the tops of the kelp fronds become exposed to the atmosphere. This is common around the coast of South Africa. Nonetheless, the elevated bromoform events with the highest mixing ratios appear to mostly occur shortly after low tide (Fig. 3). It is therefore likely that the extensive local kelp beds are an important source of the bromoform observed at the station.

**Table 1.** Selected comparison measurements of bromoform in air samples above coastal, upwelling, open ocean and lower marine boundary layer regions. Tidal height is given in m above the standard datum.

| Location | Date | Latitude | CHBr$_3$ ( ppt) | | | Reference | Region |
| | | | min | max | mean | | |
|---|---|---|---|---|---|---|---|
| New Hampshire TF | Jun-Aug 2002-2004 | 43.1 °N | 0.2 | 37.9 | 5.3-6.3 | Zhou et al., 2008 | Coastal |
| Hateruma Island, Island | Dec 2007-Nov 2008 | 24 °N | 0.5 | 7 | 0.91-1.28 | Yokouchi et al., 2017 | Coastal |
| Mauritanian upwelling | Mar-Apr 2005 | 16-21 °N | 0.1 | 0.6 | 0.2 | Quack et al., 2007b | Upwelling |
| Cape Verde | May-Jun 2007 | 16.8 °N | 2.0 | 43.7 | 4.3-13.5 | O'Brien et al., 2009 | Coastal |
| R/V Sonne | July 2014 | 2-16 °N | 0.79 | 5.07 | 2.08 | Fuhlbrügge et al., 2016a | Open ocean |
| R/A Falcon | July 2014 | 2-16 °N | 0.99 | 3.78 | 1.90 | Fuhlbrügge et al., 2016a | MABL WASP |
| Atlantic Ocean | Oct-Nov 2002 | 10 °N | 0.5 | 27.2 | - | Quack et al., 2004 | Open ocean |
| SHIVA | Nov-Dec 2011 | 0-8 °N | 1.23 | 3.35 | 1.81 | Sala et al., 2014 | MABL WASP |
| Borneo | Apr-Jul 2008 | 4.70 °N | 2-5 | 60 | - | Pyle et al., 2011 | Coastal |
| Strait of Malacca | Jun-Jul 2013 | 2-6 °N | 1.85 | 5.25 | 3.69 | Mohd Nadzir et al., 2014 | Coastal |
| Sulu-Sulawesi | Jun-Jul 2013 | 2-6 °N | 1.07 | 2.61 | 1.60 | Mohd Nadzir et al., 2014 | Coastal |
| Christmas Island | Jan 2003 | 1.98 °N | 1.1 | 31.4 | 5.6-23.8 | Yokouchi et al., 2005 | Coastal |
| San Cristobol Island | Feb-Mar 2002, 2003 | 0.92 °S | 4.2 | 43.6 | 14.2 | Yokouchi et al., 2005 | Coastal |
| Peruvian upwelling | Dec 2012 | 5-16 °S | 1.5 | 5.9 | 2.9 | Fuhlbrügge et al., 2016b | Upwelling |
| Indian ocean | Jul-Aug 2014 | 2-30 °S | 0.68 | 2.97 | 1.2 | Fiehn et al., 2017 | Open ocean |
| **Cape Point** | **Oct-Nov 2011** | **34.5 °S** | **4.4** | **64.6** | **24.8** | **This study** | **Coastal** |
| Cape Grim | 2003 | 40.7 °S | 1.3 | 6.4 | 2.9 | Yokouchi et al., 2005 | Coastal |
| Coastal South America | Dec 2007-Jan 2008 | 55 °S | 1.8 | 11 | 7.4 | Mattsson et al., 2013 | Coastal |
| Antarctic coast | Dec 2007-Jan 2008 | 65 °S | 2.1 | 4.9 | 3.2 | Mattsson et al., 2013 | Coastal |
| Antarctic Ocean | Dec 2007-Jan 2008 | 65-67 °S | 1.9 | 3.9 | 2.3 | Mattsson et al., 2013 | Open ocean |

**Air mass characterisation**

Radon ($^{222}$Rn) and CO have been extensively used as tracers for continental and anthropogenic contamination, respectively, in air mass characterisation including at Cape Point (Brunke et al., 2004). The measurements of radon ($^{222}$Rn) and carbon monoxide (CO), which were generally extremely low, show short elevated periods in the observations (Fig. 3). This indicates that majority of the bromoform measurements made at Cape Point were under clean marine conditions. Of the 1535 half hourly measurements that make up the meteorological data observed at Cape Point during October/November, 68 % were of clean marine origin. The bromoform mixing ratios in this clean air displayed a mean 23.5 ppt and ranged between 5.12 and 64.6 ppt (Table 2). The variations in $^{222}$Rn and CO concentrations occurred concurrently and mostly when the wind is from a northwesterly direction, which suggests a continental origin and therefore anthropogenic contributions to the chemical

**Table 2.** Comparison of bromoform mixing ratios from different air mass sources, sorted by radon concentration.

|  | **Clean Marine** | **Intermediate** | **Continental** |
|---|---|---|---|
| $^{222}$Rn mBq m$^{-3}$ (number) | < 350 (1028) | 800-1500 (115) | >1500 (45) |
| Mean CHBr$_3$ ppt (number of samples) | 23.5 (91) | 24.3 (12) | NA |
| Range CHBr$_3$ (ppt) | 5.1-64.6 | 4.4-46.7 | NA |

composition of the air masses. The continental contaminated air made up 9 % of the total measurements, with intermediate air masses accounting for 7.5 % of the measurements.

The bromoform mixing ratios in intermediate air samples showed a similar mean to that of clean marine air with a mean of 24.3 ppt (Table 2). The introduction of intermediate or continental air at Cape Point potentially allows for the determination of the scale of the anthropogenic contributions in general for this region. Since the intrusion of intermediate air occurs predominantly in winter, a longer time series could test the relative contributions more extensively. In the case presented here, we are not able to conclusively separate anthropogenic and biogenic sources, due to the limited, single species data set. However, given the small difference in means, the data suggests that an anthropogenic contribution is not significant. The extensive kelp beds present to the north of Cape Town further complicate the matter. An expanded suite of sampled compounds would assist in the separation of sources through the examination of related compounds such as the ratio to CH$_2$Br$_2$. It has been well documented that the contribution of anthropogenically produced bromoform is generally smaller than from natural processes on a global scale (Quack and Suess, 1999; Quack and Wallace, 2003). While, on a local scale anthropogenic source can dominate (Quack and Suess, 1999) during this sampling period the dominant contribution of bromoform was from the clean marine air masses and therefore, from biogenic sources (Table 2).

Atmospheric bromoform measurements from Mace Head, Ireland show periods of elevated mixing ratios (D. Young, pers. comm. 2017). Analysis of these elevated mixing ratios at Mace Head suggests that the local marshes may be the most likely source. However, the reason why the marshes should be a source of bromoform remains unclear at this time. Although not surrounded by marshes, Cape Point is enclosed by natural vegetation called fynbos. It is possible that the fynbos releases bromoform into the local atmosphere. This would be particularly pertinent with air masses arriving from the north. A small study has previously examined the bromoform emissions from fynbos when burnt (Kuyper et al., 2012). The measured mixing ratios in this study showed a high degree of variability with a mean of 33.9 ppt and standard deviation of 40 ppt. The limited scope and high variability meant that no firm conclusions could be drawn regarding the release of bromoform from the fynbos (Kuyper et al., 2012).

**Meteorology**

Wind speed has a complicated relationship with observed bromoform mixing ratios in marine air. The processes of bromoform sea-air flux and atmospheric dilution, both proportional to wind speed, oppose each other in their effect on the atmospheric concentration of bromoform at a given location. At low wind speeds there is a low dilution and bromoform flux into the

atmosphere. As the wind speed increases so do the rates of dilution and gas flux. The wind speed observed at Cape Point over this sampling period was dominated by lower wind speeds ($<10$ m s$^{-1}$). The full range extended from calm ($<5$ m s$^{-1}$) to occasionally reach gale force ($> 20$ m s$^{-1}$). The elevated wind speeds were associated with transient cold fronts that influence the Cape in winter and spring (Tyson and Preston-Whyte, 2000). The bromoform mixing ratios at Cape Point show a varied response to the observed wind speed; on some occasions at high wind speeds the mixing ratio was also elevated whereas at other times it was not. The lack of direct correlation at this site may be evidence of the complexity and interaction of these processes as described above.

In a coastal upwelling environment it has been shown that the height of the marine boundary layer (MBL) can play a significant role in the observed bromoform mixing ratio. For example, Fuhlbrügge et al., 2013 found that a lower marine boundary layer height acted to concentrate bromoform mixing ratios recently released from the ocean surface. Although a direct relationship between bromoform mixing ratios and MBL height was observed at Cape Point, it is possible that MBL height played a role in the measurements observed. As the MBL height is elevated the rate of atmospheric dilution increases. This would result in lower measured bromoform mixing ratios. Conversely as the MBL decreases, so the volume of atmosphere into which gases are diluted decreases, resulting in a concentrating effect and increase in measured concentration. The lack of a direct observed relationship could be a result of Cape Point sitting approximately 60 km from Cape Town international airport where the radiosondes were released. However, the effect of changes in the MBL height may be reflected in the variability of the bromoform measurements.

## 3.1 Solar radiation and Diurnal cycle

During the sampling period the solar radiation at Cape Point daily reached a level of 600-1000 W m$^{-2}$ (Fig. 3). While there was no direct correlation between solar radiation and bromoform observed, the highest mixing ratios occurred when the solar radiation was typically above 800 W m$^{-2}$.

The mean Cape Point diurnal cycle of bromoform mixing ratios displayed an increase through the morning from an estimated overnight low of 22.2 ppt, based on the first measurements of the day, to a mean maximum of 33.3 ppt at 11 pm (Fig. 4). Thereafter the mixing ratios decreased through the afternoon. A second maximum in the mean mixing ratios was observed in the early evening. This secondary maximum reached a mean mixing ratio of 34.0 ppt. There were no measurements taken between midnight and 5 am and the first morning measurements were taken prior to local sunrise. It is assumed that these measurements, taken before sunrise, were representative of the night time conditions.

This pattern in the diurnal mean bromoform mixing ratio measurements at Cape Point is similar to that observed in previously published literature (Ekdahl et al., 1998; Carpenter and Liss, 2000; Abrahamsson et al., 2004). It has been hypothesised that the increase in concentrations observed in the morning are as a result of sunrise. The onset of solar radiation stimulates photochemistry leading to oxidative stress in the kelp cells and the release of bromoform (Collén et al., 1994; Pedersén et al., 1996; Ekdahl et al., 1998). Whereas it would appear that, through this mechanism, the maxima of bromoform mixing ratios and solar radiation should coincide (Abrahamsson et al., 2004). As discussed above, changes in the height MBL act to concentrate or dilute the mixing ratio of samples in the lower atmosphere. The atmosphere into which gases can mix increases through the

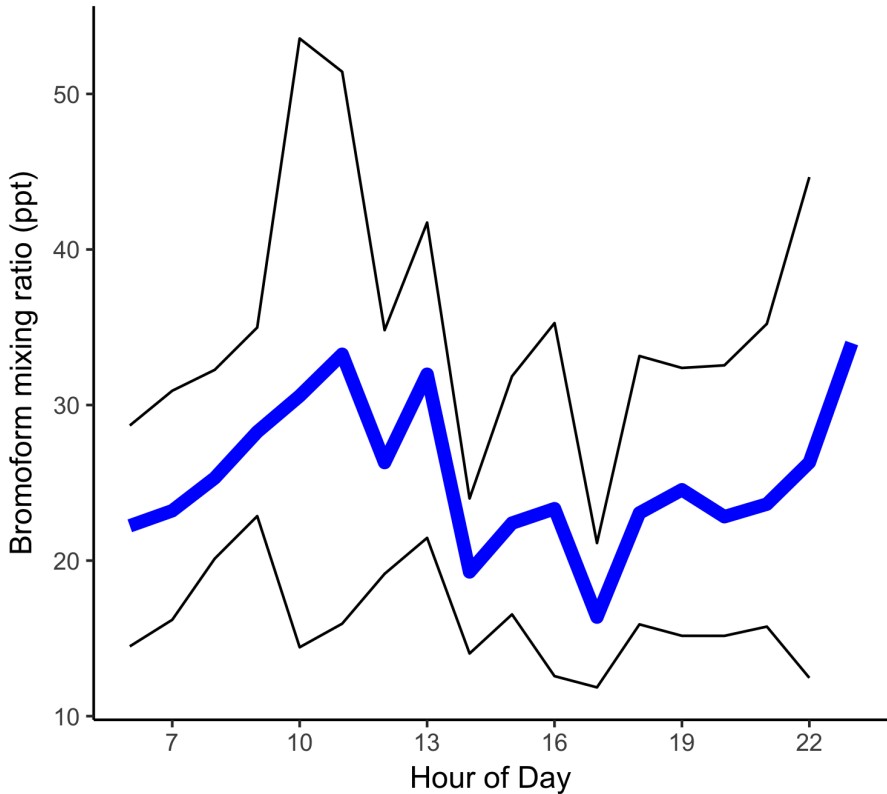

**Figure 4.** Mean diurnal cycle, calculated from all measurements binned by hour. The black lines above and below signify the 95 % confidence interval.

morning as the the MBL height rises, thus causing a dilution of trace gas in the atmosphere (Fuhlbrügge et al., 2013). This is most likely reflected in the decrease in bromoform mixing ratios at about noon and the stabalisation through the afternoon. A small contribution from the photolysis of bromoform may be present, however this would be neither detectable nor significant. A decrease in the rate of production in the afternoon or the arrival of air masses from alternate sources might explain the decrease in the late afternoon. A decreasing MBL height in the late afternoon or early evening would act to concentrate any locally released bromoform. The literature suggests that bromoform production may also be related to respiration (Ekdahl et al., 1998; Carpenter and Liss, 2000). During respiration, it is theorised that through the haloperoxidase enzyme reactions, excess intracellular hydrogen peroxide ($H_2O_2$) is removed and bromoform formed (Collén et al., 1994). Therefore, production into the evening is possible and with a lowered MBL height the measured bromoform might be large. The evening maximum in mixing ratios is, therefore, expected and consistent with previously studies in Gran Canaria and the Southern Ocean (Ekdahl et al., 1998; Abrahamsson et al., 2004).

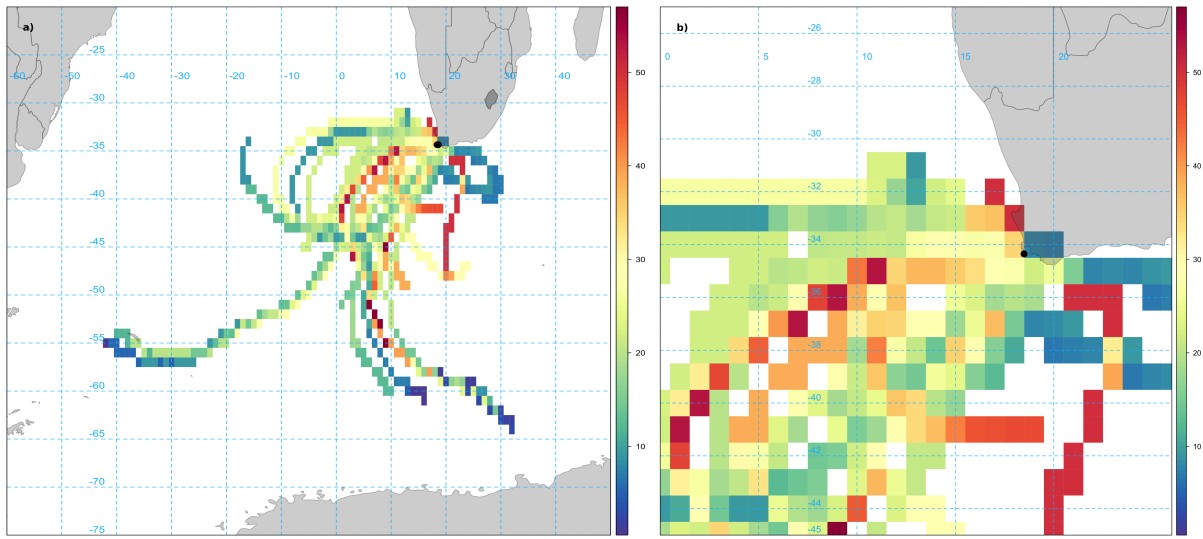

**Figure 5.** Smoothed CWT calculation of bromoform based on Hysplit back trajectories, a) Zoomed to show full extent of 96 hour back trajectories. b) Zoomed to focus on Cape Point. Units in colour bar reflect bromoform mixing ratios measured at Cape Point.

## 3.2   Back Trajectory Analysis

Given the relatively long atmospheric life time of bromoform (3 weeks), there could be sources offshore that contribute to the observed measurements at Cape Point (Carpenter and Liss, 2000). While not excluding the local source, the contribution of offshore sources was investigated using the *openair* concentration weighted trajectory (CWT) calculations of back trajectories
associated with bromoform mixing ratios. The CWT model suggests a large fetch and variability of source region for air masses arriving at Cape Point, mainly to the southwest (Fig. 5a). This large fetch included large areas of low bromoform entrainment, as would be expected from most of the open ocean, and trajectories of high entrainment. The model output suggests a number of trajectories with elevated mixing ratios. There are both off shore and inshore trajectories (5a and b). There appears to be region to the southwest of Cape Town that generates elevated mixing ratios. Furthermore, the area directly to the south of Cape
Point exhibits the highest $\bar{C}_{ij}$ values and appears to be centred over the Agulhas retroflection region (Beal et al., 2011).

The Agulhas retroflection region is an area in which the highly productive cold waters of the southern Benguela meet the warmer water from the Agulhas current. This combination of productive waters (with potentially high $CHBr_3$) and warmer sea surface temperatures driving higher rates of sea-air fluxes, could represent ideal conditions for observing higher atmospheric bromoform mixing ratios.

The the region to the southwest of South Africa extends from the coast to 45 °S and appears to contain numerous trajectories of elevated bromoform mixing ratios. It is possible that, certainly the outer areas of this region are warm core rings that have been shed off the Agulhas current. The elevated area at the coast of South Africa occurs over Cape Columbine (33 °S) into St Helena Bay (Fig. 5b). This area appears to be centred over St Helena Bay, an area known for strong coastal upwelling (Jennings

et al., 2001; Kudela et al., 2005). Given the limited nature of this data set we can not draw any firm conclusions regarding the offshore source of bromoform to Cape Point. However, this is still an interesting aspect of the region that will be monitored carefully in future work.

## 4 Conclusions

The data presented here represents the first fixed point quantitative atmospheric bromoform measurements at the Cape Point Global Atmospheric Watch Station, but also the first such dataset in southern Africa. The 135 discrete measurements made over the course of October/November 2011 exhibited a mean bromoform mixing ratio of 24.8 ± 14.8 ppt. The maximum bromoform mixing ratio reported here (64.6 ppt) was consistent with past studies, for example: that reported in Cape Verde (43.7 ppt, O'Brien et al., 2009) or New Hampshire (47.4 ppt, Zhou et al., 2008). However, it should be noted that the random errors in these measurements are quite large, with a precision of 22.2%. The scale of these uncertainties is due to the manual nature of the system, trapping to injection and oven temperature profile adjustments. Although the uncertainty associated with the data presented here is large, we feel that the data are still interesting as a first approximation of the range of values found in this region. Given the uncertainty the data should be treated with a degree of caution.

The majority of measurements (68 %) were made in clean marine air ($^{222}$Rn), implying that for these measurements the bromoform being sampled was entirely biogenic. From the data presented here it appears that the most likely source of the this bromoform is production from local kelp. Most of the periods in which bromoform concentrations were elevated for a prolonged time occurred around low tide, where kelp are exposed and most likely to produce bromoform as a response to oxidative stress. However, occasional intrusions of anthropogenically modified air may have contributed to the bromoform loading at Cape Point.

In a similar manner to the marshes surrounding Mace Head, it is possible that the fynbos vegetation at Cape Point may be a local source of bromoform to the north. The fynbos as a local source remains speculative at this stage, but will be examined going forward.

The mean diurnal pattern appears to exhibit a similar pattern to, and fall within the range of, previously published reports. An increase in the mixing ratio was observed through the morning, returning to low concentrations throughout the rest of the day. A second maximum in the mean mixing ratios was observed in the early evening. Changes in the MBL height through the day is the most likely source of variation in bromoform mixing ratios in the diurnal cycle at Cape Point.

Back trajectory analysis using the CWT model from the *openair* package provides compelling evidence to suggest an offshore biogenic source. The main region of the source appears to be centred on the Agulhas current retroflection area. A second region of elevated bromoform mixing ratios appears to exist as a transect line extending from St Helena Bay southwest off South Africa. These will be monitored carefully going forward.

Given the relatively high concentrations reported, these data indicate that this under-sampled region, may be particularly significant in terms of bromoform sources to the atmosphere. Further work needs to be done to categorise the source strength and halocarbon release from the local kelp sources. Additional measurements, both in time, space and halocarbon species, will

be required to attain a greater understanding of specific local processes governing the variability in bromoform in this region. It is thus clear that future measurements of bromoform mixing ratios at Cape Point would make an important contribution to the field. Work is currently underway to develop a more extensive halocarbon data set at Cape Point using updated equipment and calibration protocols.

*Code availability.* TEXT

*Data availability.* TEXT

*Code and data availability.* TEXT

**Appendix A**

**A1**

*Author contributions.* B. Kuyper, C. J. Palmer and C. J. C. Reason designed the experiments. B. Kuyper measured the samples of bromoform, performed analysis, wrote most of the manuscript, created all the figures. C. Labuschagne measured the meteorological and ancillary data. C. J. Palmer, C. Labuschagne and B. Kuyper performed the analysis. All authors contributed to review and improve the text.

*Competing interests.* The authors declare that they have no conflict of interest.

*Disclaimer.* TEXT

*Acknowledgements.* Funding for the development of the system was generously provided through the START Foundation. The authors would like to thank the Ernst Brunke of the South African Weather Service for his invaluable advice and support. The authors would like Mike Davies-Coleman and Dudley Shallcross for their comments and suggestions, which greatly improved the quality of the text.

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
