# Peer review of "Atmospheric bromoform at Cape Point, South Africa: An initial fixed point dataset on the African continent"

_Atmospheric Chemistry and Physics, 2017_

## Referee Comment (RC1) · Anonymous Referee #1 · 14 Jun 2017

Atmospheric bromoform at Cape Point, South Africa: a first time series on the African continent

General comments

This paper presents some in-situ atmospheric measurements (∼130 samples) of bromoform (CHBr3) from Cape Point, South Africa over a 2 month period in 2011. These are the first reported measurement of this important halocarbon from the southern African continent (although some of the data has already been discussed in Kuyper et al. 2012) so do provide a useful, if limited, new dataset. The concentrations observed are, on occasion, at the higher end of those previously reported.

[Figure]

I have a number of serious reservations about the measurements, the interpretation of the data and the conclusions which prevent me from recommending the paper for acceptance in its present form. I fully understand that sometimes it is difficult to draw firm conclusions from a limited set of measurements, but in this case I believe the authors are rather over-interpreting their data.

My major concerns are identified below, followed by some specific comments and suggestions.

1. The instrumental methods are not described particularly well. The authors refer to a previous publication which does have a lot more detail, but there are some outstanding questions related to the identification of bromoform (CHBr3) and the exclusion of possible co-eluting species. The electron capture detector (ECD) is not particularly specific so is subject to potential interferences in different types of air mass, particularly when trapping such large volumes of air or when sampling in more polluted environments. Have potential co-elutions been thoroughly tested for and ruled out?

2. Much of the discussion is highly speculative and potentially wrong. For example the conclusions on (a) anthropogenic sources of CHBr3, (b) ozone stimulation of CHBr3 release from seaweeds, and (c) a CHBr3 source from the upwelling region, are all highly dubious. The authors present very little firm evidence to support these theories and, to some extent, they overlook a more likely, or simpler, explanation for the high levels of CHBr3 at the site. They state that Cape Point and the surrounding coastlines for many kilometres to the north and west (and south?) support large, extensive seaweed beds. If this is true then surely the most likely origin of the high CHBr3 they observed is simply local (and regional) seaweed? The flux to the atmosphere would then be highly dependent on local tidal patterns, with large concentrations to be expected when the kelp becomes exposed to the air. Although this is mentioned briefly in one case study, which suggests that this process may be occurring further to the north, can the authors confirm that the same phenomena is not occurring closer to the site on a daily basis? Here a detailed description of the local /regional seaweed populations

would be very helpful, ideally identified on a map of the area (Figure 1?). The authors need to convince the reader that the high levels of CHBr3 are not simply from very local sources.

There needs to be a wider discussion of the tidal phenomenon as it is very likely to be the reason for the high concentrations and some, if not all, of the spatial/temporal variation. Similar, and more extensive, studies have been carried out over different seaweed areas such as Mace Head and NW France so references to these should also be provided.

3. Results section The order seems wrong. Why not start the results section by discussing the CHBr3 time series before going into the chemical climatologies? I would further suggest that you show the radon and wind direction data on the same plot as Figure 7 as this would make it much easier to spot patterns, etc. Similarly, when discussing the 3 individual episodes it would be helpful to see the same Figure expanded for the periods of interest.

I found some of the diagrams rather difficult to interpret. In particular the various polar plots (Figs 5, 6, 8). These types of diagram can sometimes be a little over-complicated. A better explanation, if not a full rethink, is required. For example, in the case of Figure 5, what do the individual circles represent? Are they individual samples or averages in a particular sector? Why is the red circle to the NE not represented somewhere in Figure 6. I would expect to see a red circle, albeit closer to the origin, in Figure 6. Perhaps this is because the colour scaling in the 2 Figures is different?

Why not show Figure 8 before Figs 5 and 6, perhaps after discussing the time series (see earlier comment).

However, before using these polar plots the authors need to explain why they would be expecting to see correlations of CHBr3 with tracers like CO and ozone. Furthermore, the argument that ozone increasing from, say, 25 to 35 ppb represents a significant enhancement of ozone is contentious. Can you really label this as "enhanced ozone"

and would you really expect such a small enhancement to have any significant effect on CHBr3 release from seaweeds? Please provide a reference to support this. Surely any ozone effect will be much smaller than the local tidal effect (which has not been discussed)?

There is no discussion of any diurnal pattern in the data. Were there measurements at night? Can boundary layer height or temperature have an impact on the observed concentrations?

4. The references to previous measurements (e.g. in Table 1) are not up-to-date. There have been a number of new studies in recent years that should be included (including a possible reference to the HALOCAT database).

Specific comments

P1, L2: why is the location "unique"?

P1, L13-14: the "sweet odour similar to chloroform" is irrelevant

P1, L14: what are these anthropogenic sources of bromoform? Please list with references. What fraction of global emissions are likely to be anthropogenic?

P1, L16: "Outgassing to the atmosphere" sounds better than "Atmospheric outgassing"

P2, L20: replace "within this region" with "in the tropics"

P2, L22: same as above

P2, L22: What is meant by "discrete shipboard measurements"?

P2, L23-24: "No time series …. Like Cape Point.". The authors need to be careful with this sentence. Do they mean there are no time series in Africa, the tropics or globally? The latter 2 would both be wrong. Cape Point of course is also not in the tropics.

P2, L27-28: please explain why these gases might play a significant role in climate change.

P2, L31: delete "to", i.e. "a unique location from which to measure. . .."

P2, L33: what is meant by an "intermediate air sample"?

P3, L1: Why might the subtropical location of CP make the region be an important source bromoform? Do you mean that if the region were a strong source of bromoform then this would be significant globally? I assume this is because it is relatively close to the tropics where convection could potentially transport it to the stratosphere? Please explain this sentence more clearly.

P3, L5: "biologically active" or similar is possibly better than "highly productive". Is the southern ocean active everywhere or just in certain regions? I assume the authors are referring to phytoplankton rather than macroalgae?

P3, L11: The location of the local and regional kelp beds is highly relevant to the arguments used later in the paper. Is it possible to indicate on Figure 1 where the main kelp beds are?

P4, L5: define the term "GC-ECD"

P4 L7: insert "the", i.e. "as per the method"

P4, L12: What is meant by a "quasi-discrete sampling pattern"? "Quasi-continuous would be more appropriate, although you could simply say that 131 samples were collected during the period xxx to yyy.

P4, L18: Please explain what the relevance of the sentence about gas viscosity is.

P4, L20: a ramp rate of 65 degrees per minute is very fast. Why was this necessary as it surely doesn't help with peak separation?

P4, section 2.3: there are a few details missing in this section which should be included. Was the air stream dried before pre-concentration? How did you measure the volume of air trapped? Did the system trap $CO_2$ and, if so, how did this affect the chromatography? An example chromatogram would also be helpful as well as some

discussion on possible co-elutions (see earlier comment).

P5, calibration section: this section needs some further clarification as it is not clear how the calibration was done. How does 100-300 $\mu$l of pure bromoform equate to a concentration? Was it diluted prior to trapping? What is meant by a calibration loop? How were the 99% accuracy and 12% precision estimates derived?

P5, Figure 2: Normally the fixed entity (mixing ratio) would be on the x-axis and the variable entity (peak area) would be on the y axis? What do the error bars represent (how many samples)? Why is the uncertainty given in the mixing ratio rather than the peak area?

P6, L6: "flow path" not "flow pass".

P6, L31: What is meant by "rapid shifts" on the 19th, 29th and 30th?

P6-7: what is the significance of wind speed?

P8, L7: "Measurements of bromoform at all ranges were recorded at CO levels below 100ppb". If this is the case why are there no red or orange circles in this CO range?

P8, L11: I fail to see the 2 periods of elevated ozone referred to.

P9, Fig.6: What is the impact of boundary layer height on the measurements? This might also explain some of the variability. I am not convinced that the observed variation in ozone is sufficient to be able to get any real meaning from the analysis in Fig. 6.

P9, L9: What are the stated uncertainties in the reported maximum and minimum measurements and how do these differ from the somewhat lower uncertainties in the mean?

P10, Fig.7: It would help if the 3 periods of interest were highlighted (shaded?) on the Figure. Can the authors say something about the very low values of bromoform on the Figure? There are a number of points very close to zero. Where does the air

come from at these times? Can the authors be sure that this is not a measurement problem – it seems unlikely that values would drop to zero in a region where bromoform is generally rather high?

P10, L4-5 and Fig.8: The wind speed associated with the higher concentrations to the NE and West seem very similar to me (one is described as "high" and the other as "intermediate to low"). I cannot really see any difference.

P10, L6: This sentence needs rephrasing. I assume the authors mean that at low wind speeds the average concentration was 30 ppt and they are speculating that this is maintained by some "low level" local sources? What does low level actually mean and perhaps showing wind speed in Fig. 7 would help the reader to see this more clearly.

P10, L12: "a background of low mixing ratios were observed from all wind directions". How does this relate to what was said in my previous comment? Was the "background" signal 10 ppt or 30 ppt? There are no data less than 10 ppt in the N, NE and SE sectors.

P11, episodic events: it would help to have a repeat time series for these events so the reader can see the patterns/correlations more clearly. Alternatively please put all data in one Figure (Fig 7) and perhaps some more axis markers to help distinguish between days.

P11, L11 – P12, L1: The authors state that the concentration of bromoform decreased slowly between the maximum on the 18th until the 23rd. The only problem here is that there is a large gap (several days?) in the data when we have no idea what is happening. To describe this period as a single "event" is therefore a little odd.

P12-13, Event 2: The winds are predominantly from the west-north-west but the back trajectories suggest that the air is coming from the south and east. This apparent contradiction needs to be explained. In fact the trajectories in Figure 10 for Event (b) are dated November 2011, not October. Have the authors used the correct trajectories?

P13, Event 3: again it is very difficult to follow the ups and downs in the various parameters. I find it hard to pick out 2 events in elevated ozone (line 11). Again there is a large gap with no data which makes linking the period more difficult. The authors cannot say that "bromoform rose to a maximum of 70.2 ppt" because there is a gap. P14, Section 4.1: this section is highly speculative using such terms as "circumstantial evidence" and "tentatively appear"

P14, L14-15: If air is flowing from the southern ocean in a "north-westerly direction" over Cape Town, how is it possible for this air to then pass over Cape Point which is due south of Cape Town?

P14, L18-20: there is really very little concrete evidence for this anthropogenic source and its impact on the measurements at Cape Point. There is more coast directly to the north of Cape Town so even if the air was coming from this direction and picking up anthropogenic emissions, what is to say that the bromoform and CO/radon are not coming from completely different sources?

P14, L21-22: how well do CO and Radon correlate for the entire period. It is hard to tell when they are on separate graphs using different axis ranges. It seems there are periods of high radon and low CO, but what about low radon and high CO? I assume this is unlikely if you assume a continental source for both, but it is very difficult to tell from separate figures.

P14, L30: why have you not investigated the impact of tides at the local site? (see major comment above and the next comment below).

P15, L4-5: "The extensive kelp beds at CP may contribute bromoform to both the consistent baseline and extreme events observed". If this is the case, can the presence of local kelp beds not explain the entire set of measurements? Without ruling this out, the majority of the preceding discussion is surely obsolete? Where are the local kelp beds? Are they underwater or exposed at low tide? If the latter, do you see an impact of local tide time with bromoform concentration?

P15, L6: what is meant by "other typical meteorological conditions"?

P15, L8: "quasi-continuous" is better than "quasi-discrete".

P15, L21-22, and Table 1: Why not report some median values as well as means?

P15, L25-26: the evidence for an anthropogenic source of bromoform is not really apparent (see earlier comments).

P15, L26-27: radon CO and ozone were not all elevated throughout Event 1. CO and radon were elevated at times during the period, and it is hard to say whether ozone was elevated or not. Higher ozone wouldn't necessarily be an indicator of recent anthropogenic influence. Were there no other tracers in the GC output that might help?

P16, L1-7: this section is highly speculative and rather confirms that no conclusions can be drawn as to the importance of any anthropogenic source. It would be helpful if some measurements could be made near to the water processing plants to confirm the levels of bromoform.

P16, L11: I cannot easily identify a period of "moderately elevated ozone"

P16, L12: How was the ozone "biogenic in origin"?

P16, L15: "from the west" - see point earlier about the discrepancy between measured wind direction and the back trajectories.

P16, L16: there is no clear evidence in this analysis that supports the theory of ozone-induced bromoform release.

P17, L6: the Benguela current is far to the north of CP according to Figure 1. How will this affect the concentrations at CP during Event 2? I do however agree that a study of the local kelp would be a sensible thing to do.

P17, L19: It has not been proven that the anthropogenic source of bromoform was strong during Events 1 and 2. This statement is inaccurate. In fact the whole of this

final paragraph is highly speculative.

P17, conclusions: The whole section will need to be rewritten once the various issues above have been addressed. I do also note that the extremely high values reported from Gran Canaria were measured many years before the majority of data in Table 1, so, with due respect to the original authors, I would perhaps treat these data carefully. There have been substantial improvements in analytical technology and calibration since these measurements were obtained.

---

## Referee Comment (RC2) · Anonymous Referee #2 · 7 Jul 2017

Comment on Kuyper et al., Atmospheric Bromoform at Cape Point, South Africa. . .. This manuscript discusses measurements of bromoform at a Global Atmospheric Watch station on the coast of South Africa. Coastal zones have been identified as potentially large sources of bromoform to the global atmosphere, but measurements in these regions are limited. Thus, the month long set of measurements of bromoform along the African coast is interesting and should eventually be published. However, I find myself in full agreement with the points offered by Referee #1 that the data are either over-interpreted or misinterpreted. As the authors recognize to some degree, the correlation between anthropogenic tracers (such as CO) and bromoform in certain air masses does not necessarily indicate a common source, but more likely that the sampled air masses have been exposed to multiple and independent sources. The

authors suggest that potential anthropogenic sources include water treatment plants, but this source might be readily identified by looking at the location of any nearby plant relative to Cape Point. Further, examination of the chromatograms might also reveal a different proportion of bromocarbons (e.g., dibromochloromethane/bromoform ratio) in anthropogenically influenced air vs. biogenic and kelp emissions. Without further information, I would suggest separating (or removing) the discussion of source attribution, and focus on the statistics of the bromoform measurements, including relationships to the standard GAW measurements of CO, CO2, CH4, Rn, etc. As noted by Reviewer #1, a more complete description of factors such as local and regional kelp/seaweed distributions, ocean color, tidal/diurnal factors, boundary layer height ( a significant factor for surface emissions!) would be useful in the data interpretation and discussion.

Beyond this major point, I had some additional comments and questions:

1) Regarding the title: I don't know that I would advertise a one-month campaign as a "time-series". This is especially the case, since there are large gaps in the month long data set. The measurements are sufficiently novel as "first-time" data. Also, I would not refer to the other trace gas data from the month long campaign as a "climatology".

2) Not to be too picky, but the authors suggest a great advantage for single location time series over measurements from cruises or airborne surveys. All measurements contribute to understanding the various sources and transport of trace gases. One could argue that the Cape Point site is less useful for bromoform, since it appears to be dominated by local sources. Further, though I don't argue interest in the measurements, the impact of bromoform emissions near Cape Point on stratospheric bromine is likely minimal.

3) Sampling/Analytical: I would appreciate a bit more detail on the sampling and analytical methods. For example, was there some length of inlet tubing prior to the sample trap; how was water removed prior to sample trap; were aerosols removed in any way? For the GC analysis, presumably the carrier gas was operated at constant pressure?

From the listed references, a system detection limit of 0.73 ppt bromoform is reported. This is surprisingly high for the conditions and GC system used. This DL should be included in the description since the "background" levels are only 3 x this amount. For calibration discussion, you should clarify the concentration of bromoform coming from the permeation oven. It is not 100 ul of pure bromoform. It seems more like 350 ppb of bromoform based on the flows and mixing ratios reported. Was a total of 1.5 L of air added to the trap after loading the $1 - 3$ loop injections of standard? Also, I am confused by the calibration curve and, related to that, how detector drift was calculated during the study. The peak area is determined for each known standard concentration; so the uncertainty is related to the peak area not the standard concentration. Why are the error bars associated with the known standard concentrations? Given the large uncertainty associated especially with the 3-loop standard injection (Fig, 2 and also in Kuyper, 2012 and 2014), how were intermediate detector drifts determined between samples? It seems that the individual uncertainties of a standard injection could add considerable uncertainty to the estimated drift and to the final mixing ratios reported.

4) Note that Poole, 2003 not in reference list.

5) Repeat comment of Rev. #1: the polar plots are very confusing in what they are showing. Please consider alternate plots to illustrate relationships.

6) P9, Bromoform time series. It is not clear what is the meaning of the standard deviation around the maximum and minimum (also in abstract). What is being averaged?

7) P 10. Line 1 Clarify..."the second and third events showed higher levels of bromoform compared to the first episode.

8) P11, line 9 ; high 30s ppt? should be ppb?

9) P13, fig. 10. I think Rev #1 is correct about wrong trajectories displayed for event #2. A question I have, though, is how the "event" trajectories compare to the "background" trajectories? Or if only local wind direction or 1 day trajectories are most relevant for

this site?

10) P14, line 18. As noted in my first comment, I disagree totally with this statement.

11) P15, line 8. I don't understand what this sentence means.

12) P16, line 12, What is biogenic ozone?

13) P16, Table 1, Since trajectories show potential sources from Southern Ocean, it would be informative to include data from cruises in the Southern Ocean. Plus, recent measurements have been reported from Peruvian upwelling regions (see ACP)

In summary, the manuscript by Kuyper et al. offers some interesting new measurements of bromoform from a coastal region of Africa. There may be some analytical issues with the measurements, but the data quality seems reasonable. A major revision is required, though, to simplify and rethink the data interpretation.

---

## Author Comment (AC1) · 23 Aug 2017

We would firstly like to thank the reviewer for their helpful comments. We have taken the reviewers comments into consideration and revised the manuscript accordingly. All the changes have been highlighted in the revised manuscript and are summerised as follows.

Reviewer's comments for the paper (acp-2017-244), entitled: "Atmospheric bromoform at Cape Point, South Africa, a first time series on the African continent" by Kuyper et al., submitted to ACP. Recommendation: Major revision

General comments This paper presents some in-situ atmospheric measurements (approx. 130 samples) of bromoform (CHBr3) from Cape Point, South Africa over a 2

month period in 2011. These are the first reported measurement of this important halocarbon from the southern African continent (although some of the data has already been discussed in Kuyper et al. 2012) so do provide a useful, if limited, new dataset. The concentrations observed are, on occasion, at the higher end of those previously reported. I have a number of serious reservations about the measurements, the interpretation of the data and the conclusions which prevent me from recommending the paper for acceptance in its present form. I fully understand that sometimes it is difficult to draw firm conclusions from a limited set of measurements, but in this case I believe the authors are rather over-interpreting their data.

Response: We believe that the comments and recommendations from the reviewer will greatly improve the quality and substance of this paper. The reviewer raises some concerns over the validity of the data and the conclusions. It is hoped that through these discussions / responses we can alleviate the reviewer's concerns. As the reviewer highlights it is difficult to draw firm conclusions from an inceptive dataset and limited dataset, which were on occasion at the higher end of those previously reported. In trying to bring our data to life, we have perhaps strayed into overinterpreting the data and we have tried to take on board the reviewers helpful suggestions to remedy this. As a result of the reviewer's comments, the entire calibration method was extensively re-examined and subsequently, an error identified. This has had a material impact on the data and this is reflected in the revised manuscript. Specifically, the error resulted in a systematic overestimation of the observed bromoform mixing ratios by a factor of two. Finally, the results and discussion have thus been simplified and updated as per the reviewer's suggestions.

1. The instrumental methods are not described particularly well. The authors refer to a previous publication which does have a lot more detail, but there are some outstanding questions related to the identification of bromoform ($CHBr_3$) and the exclusion of possible co-eluting species. The electron capture detector (ECD) is not particularly specific so is subject to potential interferences in different types of air mass, particularly when

trapping such large volumes of air or when sampling in more polluted environments. Have potential co-elutions been thoroughly tested for and ruled out?

Response: The reviewer is correct that the ECD is not a an absolute detection method and this is a limitation of the technique. The ECD (in combination with an analyte -specific chosen chromatographic column) is however, highly specific to polar compounds and well suited for halocarbon measurements, as has been extensively demonstrated in the literature extending back to the early 1990s (e.g. Schall and Heumann, 1993). Moreover, analyte identification by using relative retention time is a reliable and well documented method found throughout the published literature (e.g. O'Brien et al., 2009; Poole, 2003; Wevill, 2005).

The literature is also unanimous and unambiguous in showing that, in unpolluted environmental air samples, bromoform can be separated with no co-elution problems (column dependant - see below). This is made clear in the EPA 8260B method, which discusses the separation of numerous halocarbons. There is some evidence that separating bromoform from water and in highly polluted gaseous mixtures can be problematic (EPA 504.1). The co-elution problem described above is entirely column dependant; in the study presented here a DB-624 column was specifically selected for its published ability to separate bromoform without co-elution (Mattson et al., 2012; Happell Wallace,1997; Itoh et al., 1997; Andrews et al., 2016). Furthermore, Cape Point predominantly receives "clean" (i.e. with little/no detectable pollutants) marine air from the Southern Ocean and is only occasionally intruded by anthropogenically modified air from Cape Town (as is discussed in the paper). Given this the chances of co-elution would appear to be negligible for the majority of the data presented, which was recorded in pollution free air. Where the air being measured shows significant anthropogenic influence the possibility of co-elution may increase (although given the discussion above would still be expected to be low) and the discussion has been amended to reflect this potential source of uncertainty in those result which are affected. Âă

Please see below an overlay figure of a calibration standard and an air sample taken

from Cape Point. If there were co-elution occurring we would expect that the peak shape would be non-Gaussian, typically displaying tailing of the co-eluting peak. As one can see from the figure the peak shape of the calibration and the air sample for bromoform are almost identical. This gives us great confidence that the system was able to suitably separate and detect bromoform in air samples.

Remedial action: The methods section has been revised in the updated manuscript to be more explicit and unambiguous on all points raised. This includes the separation of samples as well as the calibration of GC-ECD system.

Following the reviewer's comments on co-elution, as a possible mechanism to explain the observed elevated mixing ratios, an exhaustive examination of our calibration method was performed. It was found that the trapped calibrant volume had not been correctly calculated. The permeation tube emitted pure bromoform at 373 ng min-1 and this was diluted in nitrogen at 100 ml min-1. The calibrant gas was loaded onto the trap for 30 s per sample loop. Our calculations had assumed a full minute of calibrant was loaded on the trap. Effectively a mass of 0.1865 ng of bromoform was loaded on the trap instead of the 0.373 ng previously calculated. As described in the text the number of moles of bromoform loaded in a calibration sample was calculated. This was then converted through the air number density and the number of molecules trapped to an atmospheric mixing ratio. The net effect of the correction of the error effectively resulted in a halving of the reported bromoform mixing ratios. We apologise for this error. The data has been revised as per the new calibration parameters and presented below.

Figure 1: Overlay of air sample and calibration standard. Bromoform peak indicated by the number 1. The calibration standard reflects the injection of 2 loops of bromoform standard (23.6 ppt) from the permeation oven trapped on the adsorbent trap.

As this is not intended to be a methods paper and given the method was previously published elsewhere, we do not feel that publishing this graph in ACP adds enough

to the work here to justify its inclusion. However, should the reviewer/editors deem it worthy of inclusion we would be happy to edit it and include for publication.

2. Much of the discussion is highly speculative and potentially wrong. For example the conclusions on (a) anthropogenic sources of CHBr3, (b) ozone stimulation of CHBr3 release from seaweeds, and (c) a CHBr3 source from the upwelling region, are all highly dubious. The authors present very little firm evidence to support these theories and, to some extent, they overlook a more likely, or simpler, explanation for the high levels of CHBr3 at the site. They state that Cape Point and the surrounding coast-lines for many kilometres to the north and west (and south?) support large, extensive seaweed beds. If this is true then surely the most likely origin of the high CHBr3 they observed is simply local (and regional) seaweed? The flux to the atmosphere would then be highly dependent on local tidal patterns, with large concentrations to be expected when the kelp becomes exposed to the air. Although this is mentioned briefly in one case study, which suggests that this process may be occurring further to the north, can the authors confirm that the same phenomena is not occurring closer to the site on a daily basis? Here a detailed description of the local /regional seaweed populations would be very helpful, ideally identified on a map of the area (Figure 1?). The authors need to convince the reader that the high levels of CHBr3 are not simply from very local sources.

There needs to be a wider discussion of the tidal phenomenon as it is very likely to be the reason for the high concentrations and some, if not all, of the spatial/temporal variation. Similar, and more extensive, studies have been carried out over different seaweed areas such as Mace Head and NW France so references to these should also be provided.

Response: The reviewer raises valid concerns over the discussion and the various attribution of bromoform sources. We would like to preface our response by addressing one important misconception. The three statements A-C the reviewer refers to as conclusions were not intended to be interpreted as conclusions. The purpose of A-C

is to serve as test hypotheses used to frame the discussion of our results. These were chosen as a result of (i) a careful review of the literature, to identify the sources of bromoform in past studies, and then (ii) considering this review in the context of our specific knowledge of the local environment / climate and geography, to arrive at the stated test hypothesis. We are more than aware that this first tentative data recorded at the Cape Point site does not, and could not, falsify any of the test hypothesis in and of itself. We intended to use the hypothesis as a tool to frame the discussions as to which of these potential sources of bromoform this initial data is most consistent with, with a view to guiding future research. We apologise for any ambiguity here and will aim to make this clearer.

The reviewer suggests that the local kelp may explain the measurements observed at Cape Point and that we may have omitted this possibility from our discussion. This is in fact included in the manuscript, as a it effectively a restatement of hypothesis B above. We will aim to make this clearer in the text. To reiterate what we noted above, the result do not however clearly falsify any of A-C.

The reviewer highlights that there are examples in the literature where an increase in measurement signal is associated with low tides, specifically at Mace Head (Carpenter and Liss 2000). This literature argues that this is from the fact that at low tide the kelp is exposed (entirely or partially) to the atmosphere. While the tidal range in the vertical at Cape Point is similar to that of Mace Head ($1 - 2$ m), the horizontal extent of its range is vastly different. The horizontal change in tide at Cape Point is 10 m, while at Mace Head the horizontal extent can be many 10s to 100 m (D. Shallcross, pers. comm.). Therefore, at Cape Point the kelp remain submerged throughout the tidal cycle. Thus, as the local kelp remain submerged we do not expect a large tidal signal at the Cape Point site. Work done for Kuyper (2014) showed that the bulk tide and bromoform measurements did not correlate.

In terms of the distribution and species composition of seaweed, the work of Anderson et al. (2007) divided the area in shown in the location map (Figure 1) into 19 areas

and used a variety of remote sensing technique to assess the extent of kelp beds in each of those areas. The result show that kelp beds are present in all 19 areas ranging from a minimum of 11 ha coverage in Table Bay to a maximum of just under 1000 ha towards Port Nolloth. The species composition was predominantly Ecklonia maxima Papenfuss south of Yzerfontein but transition to predominantly Laminaria pallid Greville north of Yzerfontein. Furthermore, whereas we are not excluding the local sources by any means there are arguments to support a (additional) source further afield. The bromoform signal as a function of wind speed and direction as displayed in the polar plot (Fig. 8.) suggests two sources of bromoform, one to the north and a second to the west. Although interpretation of these figures can be problematic, the figure indicates that the dominant sources are at higher wind speed thresholds and consequently from farther afield. Concurrent measurements of 222Rn and CO at Cape Point were used in an attempt to isolate sources. Some of the cases the air mass can be traced to areas influenced by anthropogenic activity, and other times not. If the wind is clean (no or very low 222Rn) and a high wind speed is present then we can assume that the bromoform is not local.

Remedial action: The discussion as a whole, along with the results, have been revised to clarify our position and further explain the points raised by the reviewer. The time series plots have been revised and includes a plot of the tide and meteorological measurements as requested. See time series figures below and the sub-plot figures. From the subplot figures it appears that the tidal height may be a necessary, but not sufficient factor to explain the elevated bromoform mixing ratios. This is explored in the revised manuscript.

3. Results section: The order seems wrong. Why not start the results section by discussing the CHBr3 time series before going into the chemical climatologies? I would further suggest that you show the radon and wind direction data on the same plot as Figure 7 as this would make it much easier to spot patterns, etc. Similarly, when discussing the 3 individual episodes it would be helpful to see the same Figure expanded

for the periods of interest.

I found some of the diagrams rather difficult to interpret. In particular the various polar plots (Figs 5, 6, 8). These types of diagram can sometimes be a little over-complicated. A better explanation, if not a full rethink, is required. For example, in the case of Figure 5, what do the individual circles represent? Are they individual samples or averages in a particular sector? Why is the red circle to the NE not represented somewhere in Figure 6. I would expect to see a red circle, albeit closer to the origin, in Figure 6. Perhaps this is because the colour scaling in the 2 Figures is different?

Why not show Figure 8 before Figs 5 and 6, perhaps after discussing the time series (see earlier comment).

However, before using these polar plots the authors need to explain why they would be expecting to see correlations of CHBr3 with tracers like CO and ozone. Furthermore, the argument that ozone increasing from, say, 25 to 35 ppb represents a significant enhancement of ozone is contentious. Can you really label this as "enhanced ozone" and would you really expect such a small enhancement to have any significant effect on CHBr3 release from seaweeds? Please provide a reference to support this. Surely any ozone effect will be much smaller than the local tidal effect (which has not been discussed)? There is no discussion of any diurnal pattern in the data. Were there measurements at night? Can boundary layer height or temperature have an impact on the observed concentrations?

Response: The reviewer expressed concern over the order and in particular some of the figures found within the results section. After consideration we concur that some of the polar plots and their interpretation proved to be problematic. Whereas we thought these figures offered valuable information, we accept the reviwers criticism that interpretation of the figures is difficult and therefore it would be better to try and tell the story without them. Thus, these the majority of these figures have been removed. As noted we now have the tracer data plotting in times series alongside bromoform

as the reviewer requested. In terms of why CO is expected to be related in some way to bromoform, it is here being used as a potential proxy for anthropogenic emissions. Bromoform has some known anthropogenic sources such as water chlorination and nuclear power generation (Quack and Wallace, 2003). While, the anthropogenic sources are on the whole small in comparison to natural sources they can, however, be significant on a local or regional scale (Quack and Suess, 1999). Since CO is a known tracer of anthropogenic influence, measurement of elevated CO in an air mass strongly suggests some type of anthropogenic mixing. A correlation of elevated CO and bromoform was expected, if the bromoform was anthropogenic in origin. The absence of a correlation being observed helps demonstrate that the bromoform measured here was not predominantly anthropogenic in origin. In terms of the reasons for measuring ozone, the presence of ozone has been shown to elicit a bromoform release in species of Laminaria kelp (Palmer et al., 2005). Palmer et al. (2005) show that ozone can elicit a significant and rapid response from kelp when ozone is introduced. Although the kelp are covered for most of the tidal range, the tops of the kelp fronds at Cape Point are exposed. So the ozone measurements were intended to help investigate whether exposure of the fronds to sunlight and ozone would notably impact the bromoform observed. Work was done in Kuyper (2014) in which these multivariate comparisons were performed. The net result was that there were too few data to conclusively explore this. The analysis was thus simplified to examine these in isolation. We thank the reviewer for bringing up the diurnal cycle as this was something of an omission and appears to provide some interesting insight.

Remedial action: The section has been rearranged and reordered to make the flow more logical as per the reviwers comments. Furthermore, most of the polar plots have been removed as they appear to be a source of confusion. Discussion about 'enhanced ozone' has been removed or revised. The reasons for investigating a correlation between bromoform and other tracers have been made clear. A discussion of the diurnal cycle and any possible impacts from temperature and marine boundary layer height have been included in the updated manuscript.

Figure 2: Time series plot of measurements at Cape Point during October / November 2011. Events are highlighted by the coloured lines: E1 = red, E2 = green, E3 = violet. Attached as figure 2.

Figure 3: Mean diurnal cycle, calculated from all measurements binned by hour. The black lines above and below signify the 95

4. The references to previous measurements (e.g. in Table 1) are not up-to-date. There have been a number of new studies in recent years that should be included (including a possible reference to the HALOCAT database).

Response: The reviewer kindly noted that the Table of observations included in the paper is not up to date. We thank the reviewer for highlighting this oversight on our part.

Remedial action: The table has been updated to include some recent measurements and reference to the HALOCAT database. The updated measurements from Cape Point have been included in this table.

Table: Selected comparison measurements of bromoform in air samples above coastal, upwelling, open ocean and lower marine boundary layer regions. Attached as Figure 7.

Specific comments

P1, L2: why is the location "unique"?

Response: Cape Point is unique in receiving clean air from the Southern Ocean and occasional influences from urban-anthropogenic sources.

Remedial action: The word unique has been removed.

P1, L13-14: the "sweet odour similar to chloroform" is irrelevant

Remedial action: This has been removed.

P1, L14: what are these anthropogenic sources of bromoform? Please list with references. What fraction of global emissions are likely to be anthropogenic?

Response: The likely anthropogenic sources in Cape Town are the nuclear power plant north of Cape Town and the numerous water treatment plants throughout the city that relies heavily on chlorination processes. From Quack and Wallace (2003) and Carpenter and Liss (2000), the estimated global budget of bromoform is total and divided into 1.6 (0.4–2.7) Gmol Br yr-1 for kelp, 2 Gmol Br yr-1 phytoplankton and 0.346 Gmol Br yr-1 anthropogenic (Quack and Wallace, 2003).

Remedial action: These have been added to the text which now reads: Bromoform, apart from the few anthropogenic sources including water chlorination, nuclear power and rice paddies, is naturally produced by kelp and phytoplankton in the upper layers of the ocean (Quack and Wallace, 2003). It is estimated that globally between 2.2 x 1011 – 2.5 x 1012 g CHBr3 yr-1i is produced of which only 3.0 x 1010 g CHBr3 yr-1 is anthropogenic, the rest being from natural sources, including 1.3 x 1011 g CHBr3 yr-1 is from brown algae and 1.7 x 1011 g CHBr3 yr-1 from phytoplankton (Quack and Wallace, 2003, Carpenter and Liss, 2000).

P1, L16: "Outgassing to the atmosphere" sounds better than "Atmospheric outgassing" P2,

Remedial action: This has been changed as requested.

L20: replace "within this region" with "in the tropics"

Remedial action: These have been replaced as requested.

P2, L22: same as above

Remedial action: These have been replaced as requested.

P2, L22: What is meant by "discrete shipboard measurements"?

Response: In this we intended to imply measurements that occur only when there are

cruises which happen to pass through the region/ area of interest. They are not fixed point measurements and have not been made over long periods of time. I.e. they are highly irregular.

Remedial action: The sentence has been revised to read: "Existing data in this region tend to be from transient ship cruises, which only provide a discrete snapshot at the point in space/time that the cruise transects the area of interest."

P2, L23-24: "No time series . . .. Like Cape Point.". The authors need to be careful with this sentence. Do they mean there are no time series in Africa, the tropics or globally? The latter 2 would both be wrong. Cape Point of course is also not in the tropics.

Response: We thank the reviewer for highlighting this potential ambiguity.

Remedial action: The sentence has been revised to be more accurate. It now reads: "No time series of measurements at a fixed point currently exists for a coastal site in southern Africa. Furthermore, the Cape Point monitoring station fills a critical Southern Hemisphere latitudinal gap between Cape Matatula, American Samoa (14 °S) and Cape Grim, Tasmania (41 °S; Brunke and Halliday, 1983)."

P2, L27-28: please explain why these gases might play a significant role in climate change.

Response: Ozone in the upper troposphere is a potent greenhouse gas and bromine initiated destruction of ozone occurs in this region.

Remedial action: An explanation of the role of ozone in the upper troposphere and it greenhouse potential in this region have been added to in the text.

P2, L31: delete "to", i.e. "a unique location from which to measure. . .."

Remedial action: This has been deleted.

P2, L33: what is meant by an "intermediate air sample"?

Response: An intermediate air sample is a marine air sample that has become modified with continental air masses (as defined in Brunke et al., 2004.). This is indicated by the radon concentration; marine air typically has a radon concentration below 350 mBq m-3, intermediate (or mixed) air samples between 1500 – 2000 mBq m-3 and continental air at above 2000 mBq m-3. mixed air contains marine, continental and urban influences (Brunke et al., 2004).

Remedial action: The text has been updated to: mixed air as defined in Brunke et al. 2004. " . . . ranging from marine sources (baseline), to continental and mixed air (baseline, continental and urban influences; Brunke et al 2004)."

P3, L1: Why might the subtropical location of CP make the region be an important source bromoform? Do you mean that if the region were a strong source of bromoform then this would be significant globally? I assume this is because it is relatively close to the tropics where convection could potentially transport it to the stratosphere? Please explain this sentence more clearly.

Response: A bit of both. That this could be a large contribution to the global budget of bromoform. Although in all likelihood the contribution from the sub-tropics is going to be small, the significance of its proximity to the tropics and the deep convection cannot be overlooked.

Remedial action: The paragraph has been revised to redress this ambiguity. It now reads: "The subtropical location of Cape Point may make this region a particularly significant source of bromoform to the atmosphere, specifically when considering potential impact on global ozone budgets; the region lies in close proximity to the tropics where deep convection is able to rapidly transport the outgassed bromoform into the UT/LS. It is here that bromine initiated catalytic ozone destruction occurs. Moreover, data recorded here is of particular value as the size of the contribution from Cape Point region is to date largely untested."

P3, L5: "biologically active" or similar is possibly better than "highly productive". Is the

Southern Ocean active everywhere or just in certain regions? I assume the authors are referring to phytoplankton rather than macroalgae?

Response: The Southern Ocean is well recognised as a high productivity low nutrient ecosystem and to a first approximation exhibits homogeneously high phytoplankton growth. There are of course fine scale various in time and space, but for the purposes of this work we suggest that the approximation of homogeneous high biological activity is sufficient. In this pelagic environment we are of course referring to microalgal productivity and not kelp.

Remedial action: The terms have been replaced as suggested by the reviewer. The text has been amended to remove any ambiguity.

P3, L11: The location of the local and regional kelp beds is highly relevant to the arguments used later in the paper. Is it possible to indicate on Figure 1 where the main kelp beds are?

Remedial action: The discussion has been modified to expand on location and type of kelp found in area surrounding Cape Point and to the north up the coast. Please also see earlier comment.

P4, L5: define the term "GC-ECD"

Remedial action: This has been done.

P4 L7: insert "the", i.e. "as per the method"

Remedial action: This has been done.

P4, L12: What is meant by a "quasi-discrete sampling pattern"? "Quasi-continuous would be more appropriate, although you could simply say that 131 samples were collected during the period xxx to yyy.

Response: Later the reviewer asks for the description of 'quasi-discrete' to be changed to 'quasi-continuous'.

Remedial action: We have followed the reviewers' suggestions here and later.

P4, L18: Please explain what the relevance of the sentence about gas viscosity is.

Response: As the temperature of a gas increases so does its viscosity and so the flow rate decreases. In modern GC systems, an electronic pressure controller regulates the pressure at the head of the column to ensure a constant flow. The simplicity of the GC-ECD used here system meant that no adjustment could be made to account for changes in gas viscosity. This can have an effect on the separation of samples. Since this was consistent through all our samples this does not pose a problem, but the authors initially thought this fact should be noted for completion sake.

Remedial action: The sentence has been removed as it appears to be causing more confusion than clarity of understanding. A statement about the gas pressure being run at constant pressure has been added to the text as follows: "Sampling

... A 30 ml min-1 nitrogen flow was added directly to the ECD in the form of make-up gas. Helium (Grade 5.0, Air Liquide) at a constant flow rate of 5 ml min-1 was maintained through the system. The oven was maintained at 35 °C for 5 min following the injection of a sample. Thereafter, the temperature was increased to 60, 90, 150, and 200 °C every 5 min. The temperature in the oven was increased at 65 °C min-1 and held isothermally once the new temperature was reached."

P4, L20: a ramp rate of 65 degrees per minute is very fast. Why was this necessary as it surely doesn't help with peak separation?

Response: The simple answer is that the GC model was not able to ramp at any other rate. A temperature could be programmed into the controller, resulting in a current being applied to the element and heating the oven. It was determined experimentally that this heating occurred at 65 °C per minute. Once each 'ramp' or increase was complete, the oven was maintained at the new temperature for a period of 5 minutes, allowing for separation at that temperature to occur.

Remedial action: The paragraph has been rewritten to remove mention to a temperature ramp. The word ramp is a technical term that is not appropriate in this context. Please see the updated text above.

P4, section 2.3: there are a few details missing in this section which should be included. Was the air stream dried before pre-concentration? How did you measure the volume of air trapped? Did the system trap CO2 and, if so, how did this affect the chromatography? An example chromatogram would also be helpful as well as some discussion on possible co-elutions (see earlier comment).

Remedial action: The text in the methods section has been revised and expanded. As per the text sample volume was calculated based on sampling rate and time. A flow rate of 100 ml min-1 through the trap was measured on a digital flow meter. Since the trap adsorbents and desiccant remained constant so did the gas flow rate. This was checked on a weekly basis. A time of 15 minutes was used to collect the sample resulting in a 1.5 l sample volume.

The trap was flushed with helium (grade 5.0) before and after trapping and adsorbents were picked specifically such that they did not retain CO2. The relevant paragraphs now read as follows: "Sampling . . .

Air samples were pre-concentrated in a custom built thermal desorption unit (TDU, Kuyper et al., 2012). Adsorbents (Carbopac X and Carboxen 1016, 9 mg each) held in a glass tube were cooled to -20 °C during the trapping phase. To exclude air from the adsorbent trap a flow of helium (100 ml min-1, Grade 5.0) was maintained both before and after sampling. Before being passed to the trap, samples were dried using magnesium perchlorate held in glass moisture trap (as per Moore and Groszko, 1999). Air was passed through the adsorbent trap at 100 ml min-1 for 15 min, resulting in a 1.5 l sample size. The sampling flow rate was checked weekly by means of a digital flow meter. The cooling of the system was achieved by a recirculating chiller filled with glycol. An oil free piston pump was used to draw air through a 60 m Decabon

sampling line and the adsorbent trap. This was routed through a T-piece with the excess gas vented to the atmosphere. A mass flow controller was used to regulate the gas flow through the adsorbent trap. The pump was operated at 400 ml min-1 and a needle valve on the exhaust was used to provide sufficient pressure for the mass flow controller to operate.

A built-in resistance wire heated the glass tube to 400 °C to desorb samples for injection. A second stage cryo-focusing system was used at the head of the column, with liquid nitrogen to improve the chromatography. The liquid nitrogen was held at the head for the duration of the primary injection. Thereafter, boiled water was used to desorb the samples trapped at the head of the column."

See above for for discussion of co-elution and note that as this is not intended to be a methods paper and given the method was previously published elsewhere, we do not feel the publishing this graph in ACP adds enough to the work here to justify its inclusion. However, should the reviewer/editors deem it is worthy of inclusion we would be happy to edit it and include for publication.

P5, calibration section: this section needs some further clarification as it is not clear how the calibration was done. How does 100-300 $\mu$l of pure bromoform equate to a concentration? Was it diluted prior to trapping? What is meant by a calibration loop? How were the 99

Response: We would like the thank the reviewer for bringing our attention to the ambiguity in this section and hope our description below is now clear. Attention has also been paid to calibrant sample flow and trapped volume. We apologise for the calibration error in the manuscript as originally submitted. The 99

Remedial action: The calibration section in the methods has been rewritten and now reads as follows: Calibration

"An external calibration method was used to verify the system performance. A custom

built permeation oven was used to deliver aliquots of bromoform at varying concentrations to the trap (Wevill and Carpenter, 2004; Kuyper, 2014). A bromoform permeation tube held at 70 °C (permeating at 343 ng min-1) was flushed with nitrogen (grade 5.0, Air Liquide) at 100 ml/min. This gas mixture was continually passed through a 100 $\mu$l sample loop and exhausted through a halocarbon trap. Aliquots of 100-300 $\mu$l (1 - 3 sample loops) of the resulting permeation gas (bromoform diluted in nitrogen), were introduced to the thermal desorption unit from the permeation oven. The sample loop was flushed for 30 s to ensure complete transport of the calibrant onto the adsorbent trap. Calibration samples were passed through the drying trap as for air samples, thus any loss would be consistent for air and calibration methods. The calibration points were analysed using on the same temperature programme as air samples to ensure identical retention times. These were also used for the identification of bromoform.

A complete calibration curve (Fig. 2) was measured prior to the start of the experimental period. The peak area was determined from the injection of 1 - 3 loops of diluted bromoform in nitrogen gas. Peak areas were calculated through the trapezoid method of integration (Poole, 2003). These areas were computed in MATLAB. The mixing ratios of the injected loops were calculated as the number of moles injected. Each loop injection resulted in 0.1865 ng of bromoform being loaded on the trap, based on the calibrated rate of the permeation tube (Weville et al. 2004; Kuyper 2014). The number of moles of bromoform on the trap was calculated. Through the air number density and the number of molecules loaded on the trap, the number of moles (bromoform) was converted to a mixing ratio. Calibration standards and air samples were run through the system independently of each other.

The variability of the peak areas measured based on repeated loop injections was converted to a 95 Thereafter, a calibration point of 1-3 loops was run every 5 air samples to account for system drift. Based on a linear regression between the introduced sample and peak area response a 99

P5, Figure 2: Normally the fixed entity (mixing ratio) would be on the x-axis and the

variable entity (peak area) would be on the y axis? What do the error bars represent (how many samples)? Why is the uncertainty given in the mixing ratio rather than the peak area?

Response: The calibration is done by the injection of a known number of loops. The number of loops is an arbitrary measure however. What is of much more interest is the measured peak area and how this relates to actual mixing ratios since the peak area is related to the concentration or mixing ratio. By controlling the number of loop injections, we effectively control the peak area so this goes on our x axis. The mixing ratios are then calculated from the measured peak areas. The uncertainty displayed through the error bars reflects variations in the measured peak area from repeated injections. An example of the calibration curve calculated in this manner can be found in Wevill and Carpenter (2004).

Remedial action: As far as we can determine this figure is correct. The text has been revised to better explain the calibration curve.

P6, L6: "flow path" not "flow pass".

Remedial action: This has been corrected in the revised manuscript.

P6, L31: What is meant by "rapid shifts" on the 19th, 29th and 30th?

Response: The rapid shifts referred to the rapid changes in wind direction.

Remedial action: The text has been amended to reflect this. "The transit of weaker cold fronts caused the occasional rapid shift in wind direction . . ."

P6-7: what is the significance of wind speed?

Response: Wind speed is important in gas measurements for a number of reasons. The bromoform sea-air gas flux rate may be approximated as a function of either the square or cube of the wind speed; e.g. Nightingale et al., (2000): k = 0.31u2(Sc/660)-1/2. The rate of atmospheric dilution also increases with wind speed. The wind speed

can also be an indication of wind fetch. These factors combined play a role in determining the observed mixing ratio at any given point in time.

Remedial action: A discussion about the role of wind speed in the variation of bromoform mixing ratios has been added to the results and discussion.

P8, L7: "Measurements of bromoform at all ranges were recorded at CO levels below 100 ppb". If this is the case why are there no red or orange circles in this CO range?

Response: One of the difficulties of polar plots is that the data are binned by wind direction and averaged. In this case the smaller measurements biased the few high measurements that occurred at below 100 ppb.

Remedial action: This figure has been removed and the sentence revised to be more accurate in the updated results section.

P8, L11: I fail to see the 2 periods of elevated ozone referred to.

Remedial action: The time series figure has been updated and annotated to highlight the event periods. The discussion regarding ozone and in particular elevated ozone have been revised.

P9, Fig.6: What is the impact of boundary layer height on the measurements? This might also explain some of the variability. I am not convinced that the observed variation in ozone is sufficient to be able to get any real meaning from the analysis in Fig. 6.

Response: The marine boundary layer height at Cape Point was calculated from twice daily radiosonde measurements made at Cape Town International airport, 60 km northeast. This is a rough approximation of the MBL at Cape Point. The height of the MBL was determined by the surface and elevated temperature inversion methods (Seibert et al., 2000; Seidel et al., 2010). The calculated heights ranged from a minimum of 91 m to a maximum of over 4000 m (Kuyper 2014).

Changes in the MBL height have in past studies been reported to influence the measured concentration of bromoform (Fuhlbrügge 2013). Despite this strong relationship reported by Fuhlbrügge et al (2013), no such relationship between bromoform and MBL at Cape Point could be established in the data presented here This could be explained due to a variety of factors. Firstly uncertainties may arise due to the approximation of MBL from Cape Town International Airport (some 60km away), as this might not be representative of conditions at Cape Point. In addition to this, if the bromoform measured at Cape Point was not locally sourced it would therefore independent of local MBL height and no effect would be expected. It could be that the MBL height as ascertained at Cape Town International airport is independent of the MBL at Cape Point and therefore not significant. At this stage we do not have enough information to separate or elaborate further on these speculations.

Remedial action: We have added information relating to the MBL at Cape Town International Airport to results and discussion sections.

P9, L9: What are the stated uncertainties in the reported maximum and minimum measurements and how do these differ from the somewhat lower uncertainties in the mean?

Response: The uncertainties were calculated as a function of the precision of the measurements. The uncertainty in the mean is the standard deviation from the calculation of this value.

Remedial action: The uncertainties of the measurements have been reported as the percentages in the text rather than the calculated values.

P10, Fig.7: It would help if the 3 periods of interest were highlighted (shaded?) on the Figure. Can the authors say something about the very low values of bromoform on the Figure? There are a number of points very close to zero. Where does the air come from at these times? Can the authors be sure that this is not a measurement problem – it seems unlikely that values would drop to zero in a region where bromoform

is generally rather high?

Response: The values reported are around 4-5x above the calculated LOD and therefore are unlikely to be an artifact. We see similar low "background" concentrations reported in the literature making these observations consistent with similar studies elsewhere. The calibration curve indicates that the system displayed a linear response to bromoform over a range of 0-40 ppt. An injection of 100 ðİđţl equates to 11.8 ppt. As to the rapid changes, yes this can be expected. We see this in other regions (Pyle et al. 2011).

Based on the original calibration data, the wind direction at Cape Point when the bromoform mixing ratios were less than 5 ppt (n = 5) was predominantly from the west to northwest (245-320°). This is entirely within the background air sector, and further confirmed by radon concentration (< 250 mBq m-3). These measurements typically occurred late in the afternoon/ early evening or in one case, before sunrise. This has been revised in the results and discussion using the updated figures and data.

Under the revised dataset there were 20 (14.8

Remedial action: The figure has been updated to include vertical lines to mark the events. See time series plot earlier.

P10, L4-5 and Fig.8: The wind speed associated with the higher concentrations to the NE and West seem very similar to me (one is described as "high" and the other as "intermediate to low"). I cannot really see any difference.

Response: The reviewer is correct in their assessment. The wording of this sentence has been updated to be correct and less ambiguous.

Remedial action: The sentence and section have been revised in the updated results section. Updated polar plots of bromoform as a function of wind direction have been added to the text. The first figure shows the full dataset, while the second two show the background (WD: 120-320 °) and non-background (WD: 320-120 °) conditions of

bromoform measurements at Cape Point.

Figure 4: Polar plots of bromoform as a function of wind speed and direction. The top figure highlights all the data, while the lower plots show background and non-background respectively. These figures give an indication of possible source directions and distance.

P10, L6: This sentence needs rephrasing. I assume the authors mean that at low wind speeds the average concentration was 30 ppt and they are speculating that this is maintained by some "low level" local sources? What does low level actually mean and perhaps showing wind speed in Fig. 7 would help the reader to see this more clearly.

Remedial action: This sentence and figure have been updated in the new results section.

P10, L12: "a background of low mixing ratios were observed from all wind directions". How does this relate to what was said in my previous comment? Was the "background" signal 10 ppt or 30 ppt? There are no data less than 10 ppt in the N, NE and SE sectors.

Response: The reviewer is correct, we had introduced ambiguity through these statements. The reviewer is correct that the binned, mean data were greater than 10 ppt in the sectors N, NE and SE. The figure has been over-interpreted by the authors; especially in light of the fact that the values represented are binned means and not the actual observations.

Remedial action: The figure and text have been revised in the updated manuscript.

P11, episodic events: it would help to have a repeat time series for these events so the reader can see the patterns/correlations more clearly. Alternatively please put all data in one Figure (Fig 7) and perhaps some more axis markers to help distinguish between days.

Remedial action: An updated figure 7 has been added to the text. This has also been annotated to highlight the three events. Time series plots of the three events have also

been added to the text in the results and discussion of the updated manuscript.

Figure: Time series sub-plots expanding the elevated bromoform event days. From left to right Events 1 - 3.

P11, L11 – P12, L1: The authors state that the concentration of bromoform decreased slowly between the maximum on the 18th until the 23rd. The only problem here is that there is a large gap (several days?) in the data when we have no idea what is happening. To describe this period as a single "event" is therefore a little odd. Response: We thank the reviewer for raising this point regarding assumptions of data over gaps.

Remedial action: The event windows have been refined to exclude the data gaps. The text has been amended to reflect that event 1 terminates with the cessation of sampling at the end of the 18th. The same applies for the third event where a data gap existed in the earlier defined window. Please also see the earlier time series plot.

P12-13, Event 2: The winds are predominantly from the west-north-west but the back trajectories suggest that the air is coming from the south and east. This apparent contradiction needs to be explained. In fact the trajectories in Figure 10 for Event (b) are dated November 2011, not October. Have the authors used the correct trajectories?

Response: The reviewer is correct, the incorrect back trajectories in figure b were inserted. The authors would like to thank the reviewer for the observation.

Remedial action: The back trajectories have been updated and corrected.

Figure: Composite daily back trajectories for the selected Events (a) E1 17 – 18 October 2011, (b) E2 25 – 27 October 2011, (c), E3 7 – 9 November 2011, (d) Background samples 23 - 24 October 2011. Trajectory heights for the events are displayed below. The colours and dates correspond respectively for each event.

P13, Event 3: again it is very difficult to follow the ups and downs in the various parameters. I find it hard to pick out 2 events in elevated ozone (line 11). Again there is a large gap with no data which makes linking the period more difficult. The authors

cannot say that "bromoform rose to a maximum of 70.2 ppt" because there is a gap.

Remedial action: The text has been revised to reflect the gaps in the bromoform data. The time series plot has been updated and annotated to highlight the events. Please see the time series figure above.

P14, Section 4.1: this section is highly speculative using such terms as "circumstantial evidence" and "tentatively appear"

Response: Given the limited data set it is not easy to be decisive. The words 'circumstantial evidence' and 'tentatively appear' are entirely appropriate in the context of the data on which they are based.

P14, L14-15: If air is flowing from the Southern Ocean in a "north-westerly direction" over Cape Town, how is it possible for this air to then pass over Cape Point which is due south of Cape Town?

Response: As a result of the Coriolis effect, anticyclones rotate in a clockwise direction in the Southern Hemisphere. A quick look at a typical synoptic chart (http://www.weathersa.co.za/observations/synoptic-charts) for a NW wind in winter in Cape Town will hopefully convince the reviewer that as the anticyclone passes south of the African continent, air flows from the Southern Ocean, over Cape Town and then down to Cape Point. These are the conditions which are typical in Cape Town winter and are those experienced in the description above.

Remedial action: The text has been revised to be more clear and accurate. The discussion has been entirely revised.

P14, L18-20: there is really very little concrete evidence for this anthropogenic source and its impact on the measurements at Cape Point. There is more coast directly to the north of Cape Town so even if the air was coming from this direction and picking up anthropogenic emissions, what is to say that the bromoform and CO/radon are not coming from completely different sources?

Response: Nothing; it is entirely possible the CHBr3 was entrained over the coast and then the CO over the city later. We intended to imply here that an Anthropogenic source could not be ruled out in these samples as the air had been influenced by human activities and will revise the wording accordingly.

Remedial action: Speculative text regarding source attribution has been removed.

P14, L21-22: how well do CO and Radon correlate for the entire period. It is hard to tell when they are on separate graphs using different axis ranges. It seems there are periods of high radon and low CO, but what about low radon and high CO? I assume this is unlikely if you assume a continental source for both, but it is very difficult to tell from separate figures.

Response: Over the sampling period CO and radon measurements agree well. There are a couple of events where elevated CO can be observed while radon is low. These brief CO 'peaks' are potentially experimental artefacts but that discussion is beyond the scope of the work presented here. Brunke et al. (2004) show the source regions of CO and radon at Cape Point, based on numerous years of measurements.

Remedial action: The time series figure has been updated to include the radon measurements as well as the CO on the same time scale. Please see the time series figure above.

P14, L30: why have you not investigated the impact of tides at the local site? (see major comment above and the next comment below).

Response: Please see the major response above. An investigation of the tides was performed. For more complete details on the tide analysis please see Kuyper, 2014. The conclusion from Kuyper (2014) were that the tides played no significant role on the variability of the measurements at Cape Point.

P15, L4-5: "The extensive kelp beds at CP may contribute bromoform to both the consistent baseline and extreme events observed". If this is the case, can the presence

of local kelp beds not explain the entire set of measurements? Without ruling this out, the majority of the preceding discussion is surely obsolete? Where are the local kelp beds? Are they underwater or exposed at low tide? If the latter, do you see an impact of local tide time with bromoform concentration?

Response: Please see the major response above. The kelp beds are not totally exposed at low tide. It is possible that emissions from the local kelp beds explain the majority of the variability. From the updated time series plots, it is suggested that the tide height may be a contributing factor in the elevated bromoform events, but not an independently sufficient factor. As was noted above the data presented here do not conclusively refute any of the test hypothesis A-C. The kelp beds, as the dominant bromoform source, may have also been overlooked on an event scale basis as the published emission rates were not sufficiently large enough to explain the observed elevated bromoform mixing ratios. Remedial action: Source attribution has been removed from the discussion. The possible tidal influence has been added to the discussion, more completely. The role of tidal height and kelp exposure as a dominant source of bromoform to the atmosphere at Cape Point has been carefully explored in the revised manuscript.

P15, L6: what is meant by "other typical meteorological conditions"?

Remedial action: We have added to the text: "…such as air temperature, pressure, rainfall and global downward radiation…"

P15, L8: "quasi-continuous" is better than "quasi-discrete".

Remedial action: As per the reviewer's suggestion this has been changed throughout the updated manuscript.

P15, L21-22, and Table 1: Why not report some median values as well as means?

Response: None of the authors of the papers from which the values were drawn report the median values of the measurements made. We have report past results exactly

as per Quack and Wallace (2003), who also report only mean values in their extensive and authoritative review.

Remedial action: The table has been updated to include more recent measurements from coastal as well as open ocean reports.

P15, L25-26: the evidence for an anthropogenic source of bromoform is not really apparent (see earlier comments).

Response: The reviewer is correct and thanked for pointing this out to us. The updated figure (time series) in the results and discussion should better highlight the evidence of an anthropogenic enhancement.

Remedial action: The results and discussion have been revised as detailed above.

P15, L26-27: radon CO and ozone were not all elevated throughout Event 1. CO and radon were elevated at times during the period, and it is hard to say whether ozone was elevated or not. Higher ozone wouldn't necessarily be an indicator of recent anthropogenic influence. Were there no other tracers in the GC output that might help? Response: The only compound calibrated for at the time of the measurements, was bromoform.

Remedial action: The discussion surrounding O3 has been revised.

P16, L1-7: this section is highly speculative and rather confirms that no conclusions can be drawn as to the importance of any anthropogenic source. It would be helpful if some measurements could be made near to the water processing plants to confirm the levels of bromoform.

Response: Conducting studies into the kinetics of the bromoform production from water chlorination and the resulting budgets to the atmosphere is a whole field of research in and of itself. This however falls beyond the scope of the work presented here which sets out to try and investigate the concentrations in unpolluted baseline air from biogenic sources. The anthropogenic contribution is discussed as a potential additional

source, and it may become an important factor in the rare circumstances we have evidence to show the air being sampled is anthropogenically influenced but, this is not the main thrust of the work. As noted above we do claim to conclude that there is an anthropogenic source. Remedial action: The discussion has been revised to focus on the relationships between the bromoform measurements and the GAW measurements in background air masses. Source attribution has been limited to speculation and this has been made clear in the revised text.

P16, L11: I cannot easily identify a period of "moderately elevated ozone"

Remedial action: The line has been removed. Ozone has no significant departures.

P16, L12: How was the ozone "biogenic in origin"?

Response: The authors were attempting to imply that the ozone was stratospheric in origin and it was not formed from anthropogenic precursors.

Remedial action: The line regarding 'biogenic in origin' has been removed.

P16, L15: "from the west" - see point earlier about the discrepancy between measured wind direction and the back trajectories.

Remedial action: We would like to thank the reviewer again for this observation and the back trajectories have been amended accordingly.

P16, L16: there is no clear evidence in this analysis that supports the theory of ozone-induced bromoform release.

Response: The authors would like to thank the reviewer for their comments on this.

Remedial action: The results and discussion have been rewritten.

P17, L6: the Benguela current is far to the north of CP according to Figure 1. How will this affect the concentrations at CP during Event 2? I do however agree that a study of the local kelp would be a sensible thing to do.

Response: The Benguela Current is not exclusively far to the North, but extends all the way down to Cape Town. Moreover, with an atmospheric lifetime of 24 days it is quite possible for bromoform formed in the Northern (and more intense) Benguela to travel to Cape Point. Under north-westerly winds air is drawn from the Benguela region over Cape Town and to Cape Point. Species composition of seaweeds in the region has been discussed above. The overinterpretation of the data may have been a function of stretching all known sources to the limit to explain the elevated mixing ratios. This should be mitigated with the revised data, which is much more inline with previous measurements.

P17, L19: It has not been proven that the anthropogenic source of bromoform was strong during Events 1 and 2. This statement is inaccurate. In fact the whole of this final paragraph is highly speculative.

Response: We agree with the reviewer. That line should not have appeared in the final version.

Remedial action: The discussion text has been entirely revised to indicate that this is our speculation and not a fact.

P17, conclusions: The whole section will need to be rewritten once the various issues above have been addressed. I do also note that the extremely high values reported from Gran Canaria were measured many years before the majority of data in Table 1, so, with due respect to the original authors, I would perhaps treat these data carefully. There have been substantial improvements in analytical technology and calibration since these measurements were obtained.

Response: The reviewer is correct that the conclusions need to be updated along with the results and discussion sections.

Response: The conclusions have been revised accordingly with the Results and Discussion sections as per the reviewer's comments and suggestions. The Gran Canaria

data will be treated with some caution.

The authors would like to again thank the reviewer for their time and effort in examining the paper. Their comments have been helpful in greatly improving the quality of the paper.

References

Anderson R. J., Rand, A., Rothman, M. D., Bolton J. J., (2007), Mapping and quantifying the South African kelp resource. African Journal of Marine Science 29(3): 369-378.

Andrews, S. J., Carpenter, L. J., Apel, E. C., Atlas, E., Donets, V., Hopkins, J. R., Hornbrook, R. S., Lewis, A. C., Lidster, R. T., Lueb, R., Minaeian, J., Navarro, M., Punjabi, S., Riemer, D., Schauffler, S. (2016). A comparison of very short lived halocarbon (VSLS) and DMS aircraft measurements in the tropical west Pacific from CAST, ATTREX and CONTRAST. Atmospheric Measurement Techniques, 9(10), 5213–5225.

Brunke, E-G. Halliday E. C. (1983). Halocarbon measurements in the Southern Hemisphere since 1977. Atmospheric Environment, 17, 4, 823826.

Brunke, E. G., Labuschagne, C., Parker, B., Scheel, H. E., Whittlestone, S. (2004). Baseline air mass selection at Cape Point, South Africa: Application of 222Rn and other filter criteria to CO2. Atmospheric Environment, 38(33), 5693–5702. https://doi.org/10.1016/j.atmosenv.2004.04.024

Carpenter, L. J., Liss, P. S. (2000). On temperate sources of bromoform and other reactive organic bromine gases. Journal of Geophysical Research, 105(D16), 20539. https://doi.org/10.1029/2000JD900242

Fiehn, A., Quack, B., Hepach, H., Fuhlbrügge, S., Tegtmeier, S., Toohey, M., Atlas, E., Krüger, K. (2017). Delivery of halogenated very short-lived substances from the west Indian Ocean to the stratosphere during the Asian summer monsoon. Atmospheric Chemistry and Physics, 17, 6723-6741.

Fuhlbrügge, S., Krüger, K., Quack, B., Atlas, E., Hepach, H., Ziska, F. (2013). Impact of the marine atmospheric boundary layer conditions on VSLS abundances in the eastern tropical and subtropical North Atlantic Ocean. Atmospheric Chemistry and Physics, 13(13), 6345–6357. https://doi.org/10.5194/acp-13-6345-2013

Fuhlbrügge, S., Quack, B., Tegtmeier, B., Atlas, E., Hepach, H., Shi, Q., Raimund, S., Krüger, K. (2016a) The contribution of oceanic halocarbons to marine and free tropospheric air over the tropical West Pacific. Atmospheric Chemistry and Physics, 16, 7569-7585.

Fuhlbrügge, S., Quack, B., Atlas, E., Fiehn, A., Hepach, H., Krüger, K. (2016b). Meteorological constraints on oceanic halocarbons above the Peruvian upwelling. Atmospheric Chemistry and Physics, 16, 12205-12217.

Happell, J. D., Wallace, D. W. R. (1997). Gravimetric preparation of gas phase working standards containing volatile halogenated compounds for oceanographic applications. Deep-Sea Research Part I: Oceanographic Research Papers, 44(9–10), 1725–1738.

Itoh, N., Tsujita, M., Ando, T., Hisatomi, G., Higashi, T. (1997). Formation and emission of monohalomethanes from marine algae. Phytochemistry, 45(1), 67–73.

Kuyper, B. (2014). An investigation into source and distribution of bromoform in the Southern African and Southern Ocean Marine boundary layer. PhD Thesis, University of Cape Town. http://open.uct.ac.za/handle/11427/8804

Kuyper, B., Labuschagne, C., Philibert, R., Moyo, N., Waldron, H., Reason, C., Palmer, C. J. (2012). Development of a simplified, cost effective GC-ECD methodology for the sensitive detection of bromoform in the troposphere. Sensors (Switzerland), 12(10), 13583–13597. https://doi.org/10.3390/s121013583

Mattson, E., Karlsson, A., Smith, W. O., Abrahamsson, K. (2012). The relationship between biophysical variables and halocarbon distributions in the waters of the Amundsen and Ross Seas, Antarctica. Marine Chemistry, 140–141, 1–9.

Mattsson, E., Karlsson, A., Katarina Abrahamsson, K. (2013). Regional sinks of bromoform in the Southern Ocean. Geophysical Research Letters, 40, 3991-3669.

Mohd Nadzir, M. S., Phang, S. M., Abas, M. R., Abdul Rahman, N., Abu Samah, A., Sturges, W. T., Oram, D. E., Mills, G. P., Leedham, E. C., Pyle, J. A., Harris, N. R. P., Robinson, A. D., Ashfold, M. J., Mead, M. I., Latif, M. T., Khan, M. F., Amiruddin, A. M., Banan, N., Hanafiah, M. M. (2014). Bromocarbons in the tropical coastal and open ocean atmosphere during the 2009 Prime Expedition Scientific Cruise (PESC-09). Atmospheric Chemistry and Physics, 14, 8137-8148.

Moore, R. M. and Groszko, W. (1999). Methyl iodide distribution in the ocean and fluxes to the atmosphere. Journal of Geophysical Research, 104(C5):11.163–11.171

Nightingale, P. D., Malin, G., Law, C. S., Watson, A. J., Liss, P. S., Liddicoat, M. I., . . . Upstill-Goddard, R. C. (2000). In situ evaluation of air-sea gas exchange parametrizations using novel conservsative and volatile tracers. Global Biogeochem. Cycles, 14(1), 373–387.

O'Brien, L. M., Harris, N. R. P., Robinson, A. D., Gostlow, B., Warwick, N. J., Yang, X., Pyle, J. A. (2009). Bromocarbons in the tropical marine boundary layer at the Cape Verde Observatory – measurements and modelling. Atmospheric Chemistry and Physics Discussion, 9, 4335–4379.

Palmer, C. J., Anders, T. L., Carpenter, L. J., Küpper, F. C., McFiggans, G. B. (2005). Iodine and halocarbon response of laminaria digitata to oxidative stress and links to atmospheric new particle production. Environmental Chemistry, 2(4), 282–290. https://doi.org/10.1071/EN05078

Poole, C. F. (2003). The Essence of Chromatography. Elsevier B.V., Sara Burgerhartsraat 25, Amsterdam, The Netherlands.

Pyle, J. A., Ashfold, M. J., Harris, N. R. P., Robinson, A. D., Warwick, N. J., Carver, G. D., . . . Ong, S. (2011). Bromoform in the tropical boundary layer of the Maritime Continent during OP3. Atmospheric Chemistry and Physics, 11(2), 529–542. https://doi.org/10.5194/acp-11-529-2011

Quack, B., Atlas, E., Petrick, G., Stroud, V., Schauffler, S. Wallace, D. W. R. (2004). Oceanic bromoform sources for the tropical atmosphere, Geophysical Research Letters, 31, L23S05.

Quack, B., Suess, E. (1999). Volatile halogenated hydrocarbons over the western Pacific between 43° and 4° N. Journal of Geophysical Research, 104(98), 1663–1678.

Quack, B., Wallace, D. W. R. (2003). Air-sea flux of bromoform: Controls, rates, and implications. Global Biogeochemical Cycles, 17(1), 1 - 27. https://doi.org/10.1029/2002GB001890

Quack, B., Peeken, I., Petrick, G., Nachtigall, K. (2007). Oceanic distribution and sources of bromoform and dibromomethane in the Mauritanian upwelling. Journal of Geophysical Research: Oceans, 112(10). https://doi.org/10.1029/2006JC003803

Sala, S., Bönisch, H., Keber, T., Oram, D. E., Mills, G., Engel, A. (2014). Deriving an atmospheric budget of total organic bromine using airborne in situ measurements from the western Pacific area during SHIVA, Atmospheric Chemistry and Physics, 14, 6903-6923.

Schall, Christian, and Klaus G. Heumann. "GC determination of volatile organoiodine and organobromine compounds in Arctic seawater and air samples." Fresenius' journal of analytical chemistry 346.6-9 (1993): 717-722.)

Seibert, P., Beyrich, F., Gryning, S. E., Joffre, S., Rasmussen, A., and Tercier, P. (2000). Review and intercomparison of operational methods for the determination of the mixing height. Atmospheric Environment, 34(7):1001–1027.

Seidel, D. J., Ao, C. O., and Li, K. (2010). Estimating climatological planetary boundary layer heights from radiosonde observations: Comparison of methods and uncertainty analysis. Journal of Geophysical Research, 115(D16):D16113.

Wevill, D. J. (2005). Atmospheric and marine measurements of volatile halogenated organic compounds in coastal and open ocean environments. PhD Thesis. University of York.

Wevill, D. J., Carpenter, L. J. (2004). Automated measurement and calibration of reactive volatile halogenated organic compounds in the atmosphere. Analyst, 129(7), 634–638. https://doi.org/10.1039/b403550j

Please also note the supplement to this comment:
https://www.atmos-chem-phys-discuss.net/acp-2017-244/acp-2017-244-AC1-supplement.pdf

[Figure]

[Figure]

**Fig. 1.** Overlay of air sample and calibration standard. Bromoform peak indicated by the number 1. The calibration standard reflects the injection of 2 loops of bromoform standard (23.6 ppt) from the permeatio

[Figure]

**Fig. 2.** Time series plot of measurements at Cape Point during October / November 2011. Events are highlighted by the coloured lines: E1 = red, E2 = green, E3 = violet.

**Fig. 3.** Mean diurnal cycle, calculated from all measurements binned by hour. The black lines above and below signify the 95 % confidence interval.

[Figure]

[Figure]

**Fig. 4.** Polar plots of bromoform as a function of wind speed and direction. The top figure highlights all the data, while the lower plots show background and non-background respectively. These figures give an

[Figure]

**Fig. 5.** Time series sub-plots expanding the elevated bromoform event days. From left to right Events 1 - 3.

[Figure]

**Fig. 6.** Composite daily back trajectories for the selected Events (a) E1 17 – 18 October 2011, (b) E2 25 – 27 October 2011, (c), E3 7 – 9 November 2011, (d) Background samples 23 - 24 October 2011. Trajectory
Table: Selected comparison measurements of bromoform in air samples above coastal, upwelling, open ocean and lower marine boundary layer regions.

| Location | Date | Latitude | CHBr₃ (ppt) | | | Reference | Region |
|---|---|---|---|---|---|---|---|
| | | | Min | Max | Mean | | |
| New Hampshire TF | Jun-Aug 2002-4 | 43.1 ºN | 0.2 | 37.9 | 5.3-6.3 | Zhou et al. 2008 | Coastal |
| New Hampshire AI | Jun-Aug 2004 | 42.9 ºN | 0.9 | 47.4 | 14.3 | Zhou et al. 2008 | Coastal |
| Hateruma Island, Island | Dec 2007 - Nov 2008 | 24 ºN | ~0.5 | 7 | 0.91-1.28 | Yokuchi et al 2017 | Coastal |
| Mauritanian upwelling | Mar - Apr 2005 | 16-21 ºN | 0.1 | 0.6 | 0.2 | Quack et al. 2007 | Upwelling |
| Cape Verde | May-Jun 2007 | 16.8 ºN | 2.0 | 43.7 | 4.3-13.5 | O'Brien et al. 2009 | Coastal |
| R/V Sonne | July 2014 | 2-16 ºN | 0.79 | 5.07 | 2.08 | Fuhlbrugger et al. 2016a | Open ocean |
| R/A Falcon | July 2014 | 2-16 ºN | 0.99 | 3.78 | 1.90 | Fulbrugger et al. 2016a | MABL WASP |
| Atlantic Ocean | Oct - Nov 2002 | 10 ºN | 0.5 | 27.2 | - | Quack et al. 2004 | Open ocean |
| SHIVA | Nov-Dec 2011 | 0-8 ºN | 1.23 | 3.35 | 1.81 | Sala et al. 2014 | MABL WASP |
| Borneo | Apr-Jul 2008 | 4.70 ºN | 2-5 | ~60 | - | Pyle et al. 2011 | Coastal |
| Strait of Malacca | Jun-Jul 2013 | 2-6 ºN | 1.85 | 5.25 | 3.69 | Mohd Nadzir et al. 2016 | Coastal |
| Sulu-Sulawesi | Jun-Jul 2013 | 2-6 ºN | 1.07 | 2.61 | 1.60 | Mohd Nadzir et al. 2016 | Coastal |
| Christmas Island | Jan 2003 | 1.98 ºN | 1.1 | 31.4 | 5.6-23.8 | Yokuchi et al. 2005 | Coastal |
| San Cristobol Island | Feb - Mar 2002, 2003 | 0.92 ºS | 4.2 | 43.6 | 14.2 | Yokuchi et al. 2005 | Coastal |
| Peruvian upwelling | Dec 2012 | 5-16 ºS | 1.5 | 5.9 | 2.9 | Fuhlbrugger et al. 2016b | Upwelling |
| Indian ocean | Jul-Aug 2014 | 2-30 ºS | 0.68 | 2.97 | 1.2 | Fiehn et al. 2017 | Open ocean |
| **Cape Point** | **Oct - Nov 2011** | **34 ºS** | **1.10** | **46.2** | **13.2** | **This study (revised)** | **Coastal** |
| Cape Grim | 2003 | 40.7 ºS | 1.3 | 6.4 | 2.9 | Yokuchi et al. 2005 | Coastal |
| Coastal South America | Dec 2007 - Jan 2008 | 55 ºS | 1.8 | 11 | 7.4 | Mattsson et al. 2013 | Coastal |
| Antarctic coast | Dec 2007 - Jan 2008 | 65 ºS | 2.1 | 4.9 | 3.2 | Mattsson et al. 2013 | Coastal |
| Antarctic Ocean | Dec 2007 - Jan 2008 | 65-67 ºS | 1.9 | 3.9 | 2.3 | Mattsson et al. 2013 | Open ocean |

**Fig. 7.** Selected comparison measurements of bromoform in air samples above coastal, up-welling, open ocean and lower marine boundary layer regions.

---

## Author Comment (AC2) · 23 Aug 2017

We would firstly like to thank the reviewer for their helpful comments. We have taken the reviewers comments into consideration and revised the manuscript accordingly. All the changes have been highlighted in the revised manuscript and are detailed as follows.

Reviewer's comments for the paper (acp-2017-244), entitled: "Atmospheric bromoform at Cape Point, South Africa, a first time series on the African continent" by Kuyper et al., submitted to ACP. Recommendation: Major revision

General comments Comment on Kuyper et al., Atmospheric Bromoform at Cape Point, South Africa.... This manuscript discusses measurements of bromoform at a Global

[Figure]

Atmospheric Watch station on the coast of South Africa. Coastal zones have been identified as potentially large sources of bromoform to the global atmosphere, but measurements in these regions are limited. Thus, the month long set of measurements of bromoform along the African coast is interesting and should eventually be published. However, I find myself in full agreement with the points offered by Referee #1 that the data are either over-interpreted or misinterpreted. As the authors recognize to some degree, the correlation between anthropogenic tracers (such as CO) and bromoform in certain air masses does not necessarily indicate a common source, but more likely that the sampled air masses have been exposed to multiple and independent sources. The authors suggest that potential anthropogenic sources include water treatment plants, but this source might be readily identified by looking at the location of any nearby plant relative to Cape Point. Further, examination of the chromatograms might also reveal a different proportion of bromocarbons (e.g., dibromochloromethane/ bromoform ratio) in anthropogenically influenced air vs. biogenic and kelp emissions. Without further information, I would suggest separating (or removing) the discussion of source attribution, and focus on the statistics of the bromoform measurements, including relationships to the standard GAW measurements of CO, $CO_2$, $CH_4$, Rn, etc. As noted by Reviewer #1, a more complete description of factors such as local and regional kelp/seaweed distributions, ocean color, tidal/diurnal factors, boundary layer height (a significant factor for surface emissions!) would be useful in the data interpretation and discussion.

Response: The comments and recommendations from the reviewer will greatly improve the quality and substance of this paper. The reviewer raises some concerns over possible over-interpretation of data and highlights that the discussion surrounding source attribution is too speculative. In the vast majority of cases we agree with the reviewer and have taken on board the criticisms, elsewhere we have clarified any ambiguities. As a result of the reviewer's comments, the calibration method was extensively examined and an error identified. This has had a material impact on the data and this is reflected in the revised manuscript. The error resulted in a systematic overestimation of the observed bromoform mixing ratios by a factor of two. We believe that the revised

data is much more inline with previously published measurements.

Finally, we accept and have followed the reviewer's comment about focusing rather on relationships to the GAW measurements. This has been done, especially, in light of the calibration error. It is hoped that through this response we can alleviate the reviewer's concerns. The revised manuscript the discussion is simplified throughout, with updated results and analysis.

Major comments

1. Regarding the title: I don't know that I would advertise a one-month campaign as a "time-series". This is especially the case, since there are large gaps in the month long data set. The measurements are sufficiently novel as "first-time" data. Also, I would not refer to the other trace gas data from the month long campaign as a "climatology".

Response: We thank the author for these comments.

Remedial action: The title and terminology in the text have been revised as follows:

" Atmospheric bromoform at Cape Point, South Africa: An initial fixed point dataset on the African continent. "

Terminology: 'Climatology' has been replaced with 'local conditions' or 'meteorological conditions' depending on the situation.

2. Not to be too picky, but the authors suggest a great advantage for single location time series over measurements from cruises or airborne surveys. All measurements contribute to understanding the various sources and transport of trace gases. One could argue that the Cape Point site is less useful for bromoform, since it appears to be dominated by local sources. Further, though I don't argue interest in the measurements, the impact of bromoform emissions near Cape Point on stratospheric bromine is likely minimal.

Response: We do not dispute that any measurements in any region are beneficial.

The argument we were attempting to put forward was that the cruises that have come past Cape Town / Cape Point have been sporadic, and tend to be focused on summer when the Southern Ocean is most accessible. A fixed sampling station in this location could be a cost effective method of addressing a large gap in our data, particularly in these winter months. Nonetheless, more ship cruises and/or airborne surveys in the area would of course be of great advantage. The reviewer is correct that this data set may be biased by local sources. A longer study possibly examining the anthropogenic sources in detail could resolve this. The fact that Cape Point is on occasion impacted by anthropogenic sources does not necessarily mean that this site is less useful. Furthermore, the GAW station at Cape Point was strategically positioned where it was as it is able to capture both the clean marine background and show local anthropogenic growth, through the different seasons and wind regimes. Moreover, the measurements made there are setup to skilfully resolve the difference between the two (Brunke et al., 2004). The impact of Cape Point bromoform on stratospheric ozone may be minimal, that is untested. It is possible in the summer months, under strong SE wind conditions, that the bromoform released may be transported to the ITCZ. We agree however that this is all speculative at this stage. The contribution to the global budget and the understanding thereof is, however, of great interest and importance. Âă

Remedial action: The text has been revised to remove ambiguity over measurements. Making note that any measurements are beneficial. The reasons for the site being of specific interest have been clarified - please see comments to reviewer 1 for more details here.

3. Sampling/Analytical: I would appreciate a bit more detail on the sampling and analytical methods. For example, was there some length of inlet tubing prior to the sample trap; how was water removed prior to sample trap; were aerosols removed in any way? For the GC analysis, presumably the carrier gas was operated at constant pressure? From the listed references, a system detection limit of 0.73 ppt bromoform is reported. This is surprisingly high for the conditions and GC system used. This DL should be

included in the description since the "background" levels are only 3 x this amount. For calibration discussion, you should clarify the concentration of bromoform coming from the permeation oven. It is not 100 ul of pure bromoform. It seems more like 350 ppb of bromoform based on the flows and mixing ratios reported. Was a total of 1.5 L of air added to the trap after loading the 1 – 3 loop injections of standard? Also, I am confused by the calibration curve and, related to that, how detector drift was calculated during the study. The peak area is determined for each known standard concentration; so the uncertainty is related to the peak area not the standard concentration. Why are the error bars associated with the known standard concentrations? Given the large uncertainty associated especially with the 3-loop standard injection (Fig, 2 and also in Kuyper, 2012 and 2014), how were intermediate detector drifts determined between samples? It seems that the individual uncertainties of a standard injection could add considerable uncertainty to the estimated drift and to the final mixing ratios reported.

Response: This has all very useful and correct thank you. It has however been dealt with in our responses to the comments of reviewer 1. A chromatogram is also included in our response to reviewer 1 for reference. In summary, the methods section has been revised to better reflect the full operating parameters and include the requested information. Calibration standards and air samples were loaded on the trap independently. If a standard was loaded no air was introduced.

Remedial action: The calibration section of the methods has been rewritten to add clarity to the aspects raised here by the reviewer. This includes information regarding the detector drifts in which a standard (1-3 loops) was analysed after every 5 samples. These were compared to other standards of similar volume. Attention has also been paid to calibrant sample flow and trapped volume. We apologise for the error caused here earlier. The sampling section in the revised manuscript reads as follows: On the sampling method:

"... A 30 ml min-1 nitrogen flow was added directly to the ECD in the form of make up gas. Helium (Grade 5.0, Air Liquide) at a constant flow rate of 5 ml min-1 was

maintained through the system. The oven was maintained at 35 °C for 5 min following the injection of a sample. Thereafter the temperature was increased to 60, 90, 150, and 200 °C every 5 min. The temperature in the oven was increased at 65 °C min-1 and held isothermally once the temperature was reached."

Air samples were pre-concentrated in a custom built thermal desorption unit (TDU, Kuyper et al., 2012). Adsorbents (Carbopac X and Carboxen 1016, 9 mg each) held in a glass tube were cooled to -20 âŮȩC during the trapping phase. To exclude air from the adsorbent trap a flow of helium (100 ml min-1, Grade 5.0) was maintained both before and after sampling. Before being passed to the adsorbent trap, samples were dried using magnesium perchlorate held in glass moisture trap (Moore and Groszko, 1999). Air was passed through the adsorbent trap at 100 ml min-1 for 15 min, resulting in a 1.5 l sample size. The sampling flow rate was checked weekly by means of a digital flow meter. The cooling of the system was achieved by a recirculating chiller filled with glycol. An oil free piston pump was used to draw air through a 60 m Decabon sampling line and the trap. This was routed through a T-piece with the excess gas vented to the atmosphere. A mass flow controller was used to regulate the gas flow through the adsorbent trap. The pump was operated at 400 ml min-1 and a needle valve on the exhaust was used to provide sufficient pressure for the mass flow controller to operate.

A built in resistance wire heated the glass tube to 400 âŮȩC to desorb samples for injection. A second stage cryo-focusing system was used at the head of the column, with liquid nitrogen to improve the chromatography. The liquid nitrogen was held at the head for the duration of the primary injection. Thereafter, boiled water was used to desorb the samples trapped at the head of the column."

On the calibration:

"An external calibration method was used to verify the system performance. A custom built permeation oven was used to deliver aliquots of bromoform at varying concentrations to the trap (Wevill and Carpenter, 2004; Kuyper, 2014). A bromoform permeation

tube held at 70 °C (permeating at 343 ng min-1) was flushed with nitrogen (grade 5.0, Air Liquide) at 100 ml/min. This gas mixture was continually passed through a 100 $\mu$l sample loop and exhausted through a halocarbon trap. Aliquots of 100-300 $\mu$l (1 - 3 sample loops) of the resulting permeation gas (bromoform diluted in nitrogen), were introduced to the thermal desorption unit from the permeation oven. The sample loop was flushed for 30 s to ensure complete transport of the calibrant onto the adsorbent trap. Calibration samples were passed through the drying trap as for air samples, thus any loss would be consistent for air and calibration methods. The calibration points were analysed using on the same temperature programme as air samples to ensure identical retention times. These were also used for the identification of bromoform.

A complete calibration curve (Fig. 2) was measured prior to the start of the experimental period. The peak area was determined from the injection of 1 - 3 loops of diluted bromoform in nitrogen gas. Peak areas were calculated through the trapezoid method of integration (Poole, 2003). These areas were computed in MATLAB. The mixing ratios of the injected loops were calculated as the number of moles injected. Each loop injection resulted in 0.1865 ng of bromoform being loaded on the trap, based on the calibrated rate of the permeation tube (Weville et al. 2004; Kuyper 2014). The number of moles of bromoform on the trap was calculated. Through the air number density and the number of molecules loaded on the trap, the number of moles (bromoform) was converted to a mixing ratio. Calibration standards and air samples were run through the system independently of each other.

The variability of the peak areas measured based on repeated loop injections was converted to a 95 % confidence interval. This confidence interval was used to show the uncertainty in the conversion of measured peak area to mixing ratio. Since the peak area is proportional to the concentration in the sample, the measured peak area is controlled through the number of injected loops and thus calculated against mixing ratio (Fig. 2). Thereafter, a calibration point of 1-3 loops was run every 5 air samples to account for system drift. Based on a linear regression between the introduced sample

and peak area response a 99 % accuracy was achieved on this system. Analysis from repeated 2 loop injections indicated a system precision of 7.4 %. Following an analysis of the calibration curve a limit of detection of 0.21 ppt was determined for this system. "

4. Note that Poole, 2003 not in reference list.

Response: We would like to thank the reviewer for this observation and like to apologise for the oversight.

Remedial action: This reference has been added to the reference list.

Poole, C. F. (2003). The Essence of Chromatography. Elsevier B.V., Sara Burgerhart-sraat 25, Amsterdam, The Netherlands.

5. Repeat comment of Rev. #1: the polar plots are very confusing in what they are showing. Please consider alternate plots to illustrate relationships.

Response: We thank the reviewer for this comment - it does appear that this plots caused much confusion.

Remedial action: The results have been presented differently as suggested by the reviewer. The majority of the polar plots have been removed. A revised time series plot has been added to the results as the main focus point. The revised results also includes a diurnal variation plot, time series plots of the bromoform events and a single polar plot of bromoform as a function of wind speed and direction, as the authors felt that this figure was still instructive.

Figure 1: Time series plot of measurements at Cape Point during October / November 2011. Events are highlighted by the coloured lines: E1 = red, E2 = green, E3 = violet.

Figure 2: Mean diurnal cycle, calculated from all measurements binned by hour. The black lines above and below signify the 95 % confidence interval.

Figure3: Polar plots of bromoform as a function of wind speed and direction. The

top figure highlights all the data, while the lower plots show background and non-background respectively. These figures give an indication of possible source directions and distance.

Figure 4: Time series sub-plots expanding the elevated bromoform event days. From left to right Events 1 - 3.

6. P9, Bromoform time series. It is not clear what is the meaning of the standard deviation around the maximum and minimum (also in abstract). What is being averaged?

Response: The 'standard deviation' reported in the text regarding certain measurements are a description of uncertainty based on the precision of the instrument.

Remedial action: The wording has been revised to be clearer. Where precision uncertainty of a measurement is reported in the text the percentage is now reported rather than the calculated value.

7. P 10. Line 1 Clarify. . ."the second and third events showed higher levels of bromoform compared to the first episode.

Response: We thank the reviewer for pointing out the ambiguity of the statement. The maximum (and not specifically average) mixing ratios of bromoform were larger in events 2 and 3 than in event 1.

Remedial action: The results and discussion sections have been revised in light of this. Ăă

8. P11, line 9 ; high 30s ppt? should be ppb?

Response: We would like to thank the reviewer for noticing this. The reviewer is correct and it should be ppb.

Remedial action: This has been corrected.

9. P13, fig. 10. I think Rev #1 is correct about wrong trajectories displayed for event #2.

A question I have, though, is how the "event" trajectories compare to the "background" trajectories? Or if only local wind direction or 1 day trajectories are most relevant for this site?

Response: The reviewer is correct that the wrong back trajectory had been inserted in error into the figure. From the revised figures (below) one can see some variability in the back trajectories. However, we do not feel that the variability is not to such an extent that 1 day trajectories are necessary.

Remedial action: The back trajectories have been corrected. In response to the reviewer's comment about background trajectories, a background trajectory has been added to the figure. For reference.

Figure 5: Composite daily back trajectories for the selected Events (a) E1 17 – 18 October 2011, (b) E2 25 – 27 October 2011, (c), E3 7 – 9 November 2011, (d) Background samples 23 - 24 October 2011. Trajectory heights for the events are displayed below. The colours and dates correspond respectively for each event.

10. P14, line 18. As noted in my first comment, I disagree totally with this statement. Remedial action: The discussion has been revised as suggested by the reviewer and reviewer 1 to remove discussion about source attribution.

11. P15, line 8. I don't understand what this sentence means.

Response: We thank the reviewer for their observation regarding this sentence. That line should not have appeared in the final version, our sincere apologies.

Remedial action: The sentence has been removed in the revised discussion. Âă

12. P16, line 12, What is biogenic ozone?

Response: What was meant by this term was ozone formed in the stratosphere and not from anthropogenic precursors.

Remedial action: This has been amended accordingly in the revised results and dis-

cussion.

13. P16, Table 1, Since trajectories show potential sources from Southern Ocean, it would be informative to include data from cruises in the Southern Ocean. Plus, recent measurements have been reported from Peruvian upwelling regions (see ACP)

Response: The authors would like to thank the reviewer for this useful and insightful comment and suggestion.

Remedial action: The Table has been updated including cruises from the Southern Ocean.

Table: Selected comparison measurements of bromoform in air samples above coastal, upwelling, open ocean and lower marine boundary layer regions. The authors would like to the reviewers for their time, efforts and comments which have helped to greatly improve the substance and quality of the paper. As Figure 7.

References

Brunke, E. G., Labuschagne, C., Parker, B., Scheel, H. E., & Whittlestone, S. (2004). Baseline air mass selection at Cape Point, South Africa: Application of 222Rn and other filter criteria to CO2. Atmospheric Environment, 38(33), 5693–5702. https://doi.org/10.1016/j.atmosenv.2004.04.024

Fiehn, A., Quack, B., Hepach, H., Fuhlbrügge, S., Tegtmeier, S., Toohey, M., Atlas, E., & Krüger, K. (2017). Delivery of halogenated very short-lived substances from the west Indian Ocean to the stratosphere during the Asian summer monsoon. Atmospheric Chemistry and Physics, 17, 6723-6741.

Fuhlbrügge, S., Quack, B., Tegtmeier, B., Atlas, E., Hepach, H., Shi, Q., Raimund, S., & Krüger, K. (2016a) The contribution of oceanic halocarbons to marine and free tropospheric air over the tropical West Pacific. Atmospheric Chemistry and Physics, 16, 7569-7585.
Fuhlbrügge, S., Quack, B., Atlas, E., Fiehn, A., Hepach, H., & Krüger, K. (2016b). Meteorological constraints on oceanic halocarbons above the Peruvian upwelling. Atmospheric Chemistry and Physics, 16, 12205-12217.

Kuyper, B. (2014). An investigation into source and distribution of bromoform in the Southern African and Southern Ocean Marine boundary layer. PhD Thesis, University of Cape Town. http://open.uct.ac.za/handle/11427/8804

Mattsson, E., Karlsson, A., & Katarina Abrahamsson, K. (2013). Regional sinks of bromoform in the Southern Ocean. Geophysical Research Letters, 40, 3991-3669.

Mohd Nadzir, M. S., Phang, S. M., Abas, M. R., Abdul Rahman, N., Abu Samah, A., Sturges, W. T., Oram, D. E., Mills, G. P., Leedham, E. C., Pyle, J. A., Harris, N. R. P., Robinson, A. D., Ashfold, M. J., Mead, M. I., Latif, M. T., Khan, M. F., Amiruddin, A. M., Banan, N., & Hanafiah, M. M. (2014). Bromocarbons in the tropical coastal and open ocean atmosphere during the 2009 Prime Expedition Scientific Cruise (PESC-09). Atmospheric Chemistry and Physics, 14, 8137-8148.

O'Brien, L. M., Harris, N. R. P., Robinson, A. D., Gostlow, B., Warwick, N. J., Yang, X., & Pyle, J. A. (2009). Bromocarbons in the tropical marine boundary layer at the Cape Verde Observatory – measurements and modelling. Atmospheric Chemistry and Physics Discussion, 9, 4335–4379.

Poole, C. F. (2003). The Essence of Chromatography. Elsevier B.V., Sara Burgerhart-sraat 25, Amsterdam, The Netherlands.

Pyle, J. A., Ashfold, M. J., Harris, N. R. P., Robinson, A. D., Warwick, N. J., Carver, G. D., ... Ong, S. (2011). Bromoform in the tropical boundary layer of the Maritime Continent during OP3. Atmospheric Chemistry and Physics, 11(2), 529–542. https://doi.org/10.5194/acp-11-529-2011.

Quack, B., Atlas, E., Petrick, G., Stroud, V., Schauffler, S. & Wallace, D. W. R. (2004). Oceanic bromoform sources for the tropical atmosphere, Geophysical Research Letters, 31, L23S05.

Quack, B., Peeken, I., Petrick, G., & Nachtigall, K. (2007). Oceanic distribution and sources of bromoform and dibromomethane in the Mauritanian upwelling. Journal of Geophysical Research: Oceans, 112(10). https://doi.org/10.1029/2006JC003803

Sala, S., Bönisch, H., Keber, T., Oram, D. E., Mills, G., & Engel, A. (2014). Deriving an atmospheric budget of total organic bromine using airborne in situ measurements from the western Pacific area during SHIVA, Atmospheric Chemistry and Physics, 14, 6903-6923.

Wevill, D. J., & Carpenter, L. J. (2004). Automated measurement and calibration of reactive volatile halogenated organic compounds in the atmosphere. Analyst, 129(7), 634–638. https://doi.org/10.1039/b403550j.

Yokouchi, Y., Hasebe, F., Fujiwara, M., Takashima, H., Shiotani, M., Nishi, N., . . . Nojiri, Y. (2005). Correlations and emission ratios among bromoform, dibromochloromethane, and dibromomethane in the atmosphere. Journal of Geophysical Research Atmospheres, 110(23), 1–9. https://doi.org/10.1029/2005JD006303.

Yokouchi, Y., Saito, T., Zeng, J., Mukai, H., & Montzka, S. (2017). Seasonal variation of bromocarbons at Hateruma Island, Japan: implications for global sources. Journal of Atmospheric Chemistry, 74(2), 171–185. https://doi.org/10.1007/s10874-016-9333-9.

Zhou, Y., Mao, H., Russo, R. S., Blake, D. R., Wingenter, O. W., Haase, K. B., Ambrose, J., Varner, R. K., Talbot, R., Sive, B. C. (2008). Bromoform and dibromomethane measurements in the seacoast region of New Hampshire, 2002-2004. Journal of Geophysical Research Atmospheres, 113(8), 2002–2004. https://doi.org/10.1029/2007JD009103.

Please also note the supplement to this comment:
https://www.atmos-chem-phys-discuss.net/acp-2017-244/acp-2017-244-AC2-
supplement.pdf

[Figure]

**Fig. 1.** Time series plot of measurements at Cape Point during October / November 2011. Events are highlighted by the coloured lines: E1 = red, E2 = green, E3 = violet.

Bromoform mixing ratio (ppt)

Hour of Day

**Fig. 2.** Mean diurnal cycle, calculated from all measurements binned by hour. The black lines above and below signify the 95 % confidence interval.

[Figure]

[Figure]

**Fig. 3.** Polar plots of bromoform as a function of wind speed and direction. The top figure highlights all the data, while the lower plots show background and non-background respectively. These figures give an

[Figure]

**Fig. 4.** Time series sub-plots expanding the elevated bromoform event days. From left to right Events 1 - 3.

[Figure]

**Fig. 5.** Composite daily back trajectories for the selected Events (a) E1 17 – 18 October 2011, (b) E2 25 – 27 October 2011, (c), E3 7 – 9 November 2011, (d) Background samples 23 - 24 October 2011. Trajectory

Table: Selected comparison measurements of bromoform in air samples above coastal, upwelling, open ocean and lower marine boundary layer regions.

| Location | Date | Latitude | Min | Max | Mean | Reference | Region |
|---|---|---|---|---|---|---|---|
| | | | \multicolumn{3}{CHBr$_3$ (ppt)} | | |
| New Hampshire TF | Jun-Aug 2002-4 | 43.1 ºN | 0.2 | 37.9 | 5.3-6.3 | Zhou et al. 2008 | Coastal |
| New Hampshire AI | Jun-Aug 2004 | 42.9 ºN | 0.9 | 47.4 | 14.3 | Zhou et al. 2008 | Coastal |
| Hateruma Island, Island | Dec 2007 - Nov 2008 | 24 ºN | ~0.5 | 7 | 0.91-1.28 | Yokuchi et al 2017 | Coastal |
| Mauritanian upwelling | Mar - Apr 2005 | 16-21 ºN | 0.1 | 0.6 | 0.2 | Quack et al. 2007 | Upwelling |
| Cape Verde | May-Jun 2007 | 16.8 ºN | 2.0 | 43.7 | 4.3-13.5 | O'Brien et al. 2009 | Coastal |
| R/V Sonne | July 2014 | 2-16 ºN | 0.79 | 5.07 | 2.08 | Fuhlbrugger et al. 2016a | Open ocean |
| R/A Falcon | July 2014 | 2-16 ºN | 0.99 | 3.78 | 1.90 | Fulbrugger et al. 2016a | MABL WASP |
| Atlantic Ocean | Oct - Nov 2002 | 10 ºN | 0.5 | 27.2 | - | Quack et al. 2004 | Open ocean |
| SHIVA | Nov-Dec 2011 | 0-8 ºN | 1.23 | 3.35 | 1.81 | Sala et al. 2014 | MABL WASP |
| Borneo | Apr-Jul 2008 | 4.70 ºN | 2-5 | ~60 | - | Pyle et al. 2011 | Coastal |
| Strait of Malacca | Jun-Jul 2013 | 2-6 ºN | 1.85 | 5.25 | 3.69 | Mohd Nadzir et al. 2016 | Coastal |
| Sulu-Sulawesi | Jun-Jul 2013 | 2-6 ºN | 1.07 | 2.61 | 1.60 | Mohd Nadzir et al. 2016 | Coastal |
| Christmas Island | Jan 2003 | 1.98 ºN | 1.1 | 31.4 | 5.6-23.8 | Yokuchi et al. 2005 | Coastal |
| San Cristobol Island | Feb - Mar 2002, 2003 | 0.92 ºS | 4.2 | 43.6 | 14.2 | Yokuchi et al. 2005 | Coastal |
| Peruvian upwelling | Dec 2012 | 5-16 ºS | 1.5 | 5.9 | 2.9 | Fuhlbrugger et al. 2016b | Upwelling |
| Indian ocean | Jul-Aug 2014 | 2-30 ºS | 0.68 | 2.97 | 1.2 | Fiehn et al. 2017 | Open ocean |
| **Cape Point** | **Oct - Nov 2011** | **34 ºS** | **1.10** | **46.2** | **13.2** | **This study (revised)** | **Coastal** |
| Cape Grim | 2003 | 40.7 ºS | 1.3 | 6.4 | 2.9 | Yokuchi et al. 2005 | Coastal |
| Coastal South America | Dec 2007 - Jan 2008 | 55 ºS | 1.8 | 11 | 7.4 | Mattsson et al. 2013 | Coastal |
| Antarctic coast | Dec 2007 - Jan 2008 | 65 ºS | 2.1 | 4.9 | 3.2 | Mattsson et al. 2013 | Coastal |
| Antarctic Ocean | Dec 2007 - Jan 2008 | 65-67 ºS | 1.9 | 3.9 | 2.3 | Mattsson et al. 2013 | Open ocean |

**Fig. 6.** Selected comparison measurements of bromoform in air samples above coastal, up-welling, open ocean and lower marine boundary layer regions.

---

## Referee Report (RR1)

**Atmospheric bromoform at Cape Point… Kuyper et al. (acp-2017-244)**

Review of resubmission

The manuscript is much improved and I commend the authors on their hard work and perseverance. This still represents a rather limited dataset, but the authors have, in my opinion, made sensible alterations to their interpretation of the data and to the conclusions drawn. I still have some reservations, particular in the interpretation of some of the case studies, but I would be happy to recommend publication in ACP once the issues outlined below have been addressed.

P1, L5: replace "an" with "a"

P1, L12: apart from a few minor anthropogenic sources ….

P1, L12: are there individual references for the different anthropogenic sources or do they all come from Quack and Wallace (2003)?

P2, L30: higher atmospheric levels are not just seen in "tropical" regions – they are often associated with seaweeds at mid-latitudes as well (Mace Head, etc).

P3, L20: replace full stop with a comma – i.e. ".. into the UTLS, where bromine-initiated …"

P3, L21: add "the". i.e. " … contribution from the Cape Point region is ….".

P3, L31: delete second use of the word "extensive"

P4, L11: typo "Perkin Elmer", not "Perkin Elmar".

P8, L13: what does pm mean (13.2 pm 9.7 ppt)? plus/minus?

P8, L13: Figure number is missing

P8, L14: I don't particularly like this sentence "The measurements were largely consistent within a few days, however could vary by 10s of ppt between days". What are the authors trying to say here? Please try rephrasing. Something like "Bromoform was typically in the range of 1-20 ppt but on several occasions elevated mixing ratios were encountered that could last for several hours ….."

P8, L26 Should "Cape Town" read "Cape Point"?

P8, L28: missing parentheses around the reference.

P9, Fig 3: Units are missing from the y-axis

P9, Fig.3: please make it clearer either in the Figure caption or perhaps in the text which is high tide and which is low tide. The graph (2nd panel down) varies from 0.6 to 1.4, but what does this mean?

P9, L4-6: You say that the tidal height is a "necessary but not sufficient factor" in the high bromoform episodes but then go on to say that "it is therefore likely that the extensive local kelp beds are an important source of the observed bromoform". These statements seem to contradict themselves a little. Please consider rephrasing these 2 sentences.

It is interesting that the seaweeds do not become completely exposed at Cape Point. Is this true of the wider region as well? Are you able to smell the seaweeds at low tide? This might be a good indicator of very local emissions!

P10, L3: replace "particularly" with "including"

P11, L1: missing Table number

P11, L2-4: I don't like this sentence very much either. "The introduction of ........". How does it allow for the determination of scale of the anthropogenic contributions in this region? Do you mean anthropogenic bromoform? As I mentioned in my first review when you have seaweed beds to the north of Cape Town it is very hard to distinguish whether the CHBr3 comes from an anthropogenic source rather than a marine source further to the north.

P11, L9: strictly speaking the kelp beds were not "responsible" for the measurements. Please rephrase.

P11, L24-25: although you say there was no correlation between bromoform and boundary layer height you cannot say that BL ht has no influence. Even in the diurnal cycle shown in Figure 4, the gradually declining concentrations after 10 am could partly be due to an expanding boundary layer as the atmosphere warms up. This would cause a dilution and therefore contribute to the decline. Similarly in the evening when the nocturnal boundary forms, might this not contribute to the higher concentrations you observe in the early evening and through to 11 pm. There could also be a link between boundary layer height and tide. Low tide and low BL could lead to higher concentrations particularly if there are emissions at night?

P12, L5: delete "in"

P12, L6: the overnight low looks more like 12-13 ppt from Figure 4

P12, Fig 4: it would be useful to have an indication as to the number of samples in each hourly bin

P13, para 1: note my point about BL height (above).

P15, Fig 6: Does the sample with the maximum concentration of bromoform correspond to the back-trajectory that passes through Koeberg?

P15, Fig 6: why do the trajectories start at an altitude of over 200m? Is this the altitude of the sampling location (I had assumed it was lower)? Also, the trajectories look a bit odd as they seem to go to almost negative altitudes, particularly on the 18th? You do not comment on these altitude profiles in the text. What do they actually tell us?

P16, Fig 7: the line for mbl looks a little odd. Was there no change in boundary layer height on the 26 October? Also, the units are missing from the y-axis (same for Figs 5 and 8).

P17, L7: "The" should read "the" (no capital required).

P17, L25: missing word "between 3 and 6 pm"?

P18, L5: delete the first "known", and, better still, replace "known" with "potential". You haven't confirmed in this work that the power and water plants in CT actually produce bromoform.

P18, Fig 8: again the boundary layer height looks strange. No change between 9 am on the 7th Nov and 9 am on 8th Nov?

P19, Fig 9: The wrong date is used in the figure caption.

**Comment on the case studies**

I understand that the authors are trying to highlight some of the more interesting features in their data, but I worry that they do not really have enough data to come to any conclusions. For example,

in Case 1, the argument for an anthropogenic source is essentially based on one single data point, which occurs during a period of elevated CO and radon (and also at low tide). I wonder if the trajectories in Figure 6 could be coloured differently to show the gradual change of air mass origin over the period.

In Case 2 it is very hard to discern anything meaningful from the various parameters discussed, particularly in regard to the tidal heights. I do however notice that the wind speed increases over the period. As the winds are coming from the west, would an open ocean source (influenced by increasing winds) not be a possibility as well? Again this would be highly speculative.

---

## Author Response (AR2)

**Authors response to the reviewer comments**

**REVIEWER 1**

**Suggestions for revision or reasons for rejection (will be published if the paper is accepted for final publication)**

*The revised manuscript of Kuyper et al. has addressed many of the issues raised in the reviews. However, there are still some points that need clarification or correction.*

**Response:** The authors would like to thank the reviewer for their time in reviewing our paper. Their comments and suggestions have made a significant improvement in the quality and substance of the paper.

1) *General comment. Be sure to clearly identify all components of tables and figures in the associated captions.*

**Response:** The authors would like to thank the reviewer for this comment. We apologies for this oversight in the preparation of our figures.

**Remedial action:** The units in the figures have been added, where possible in the plot otherwise in the caption.

2) *Significance of Cape Point site to tropical convection….The authors make a reasonable claim that it is important to understand the sources and budget of bromoform in the global and regional atmospheres. However, if they claim that the emissions near Cape Point are entrained in tropical convection, I'd appreciate some reference that shows the significance of this transport path from the S. African MBL to the tropical UTLS.*

**Response:** NOAA Hysplit trajectories were run in the forward mode for 2011. These indicate that 25 % of the air masses that transition Cape Point arrive in the tropics within 96 hours. Furthermore, Tyson and Preston-Whyte (2000), show the mass transports over southern Africa. They indicate particularly in spring and autumn 25 % of the transport over southern Africa is to the Atlantic and the tropics. In summer and winter this can drop to as low as 5 %.

**Remedial action:** The following has been added to the text in the significance section:

"The seasonal synoptic conditions over South Africa results in varying transport patterns (Tyson and Preston-Whyte, 2000). During summer, approximately 5 % of trajectories from South Africa escape to the Atlantic (10 ºS), while 75 % of transport exits to the southeast. Ridging high pressure systems, present during spring and autumn, increase the transport to the Atlantic to 25 % (Tyson and Preston-Whyte, 2000)."

3) *Calibrations. While the authors have expanded on their discussion of analytical protocols and calibration, I still remain confused on some basic operations. First, the authors describe using sample loops for calibrating the system, as is a common practice. However, they then also describe an error associated with timing of the standard addition to the system, e.g., 30 sec instead of 1 minute for standards. If the*

*standards are injected via single or multiple fixed volume loop injections of the output of the permeation tube, how does timing make any difference? I am confused. Second, I do not understand the relationship between the calibration curve shown in the figure and the use of intermittent standards to track drift in system response. Presumably, response factors for bromoform are calculated regularly with runs of different amounts of standard. How these different standards (a choice of from 1 – 3 standards) are used is not at all clear. How is drift calculated, and how does that relate to the calibration curve, which I have to repeat, is not correct as shown (no matter if it was used by Wevill and Carpenter). Please refer to most any textbook on analytical chemistry. Based on the calibration curve shown, I find it hard to believe that the error bars represent the 7% uncertainty as claimed in the author response.*

**Response:** The reviewer is correct in both their points regarding the calibration. We apologise for any confusion that has been caused. The volume of the sample loop is fixed and not related to the injection time. This has been corrected in the manuscript. Furthermore, the reviewer is correct, that we should have plotted bromoform mixing ratio (independent variable) against peak area (dependant variable).

**Remedial action:** The sample loop volume in the calculation has been amended to correctly reflect the 100 µl sample volume. The calibration curve has been corrected to reflect the bromoform mixing ratio as the independent variable. The calibration text has also been revised to reflect the full calculation method:

"The mixing ratios of the injected loops were calculated from the number of moles of bromoform injected, as follows. Each loop injection resulted in 0.0343 ng of bromoform being loaded on the trap, based on the calibrated rate of the permeation tube (Wevill2004, Kuyper2014). The number of moles of bromoform on the trap was calculated from this mass which resulted in $1.36 \times 10^{-12}$ mol being loaded on the trap, per sample loop injection. The number of molecules of bromoform was calculated by multiplying the number of moles by the Avogadro constant to yield the number of bromoform molecules on the trap. The total number of molecules in a sample was calculated by multiplying the air number density ($2.5 \times 10^{25}$) with the sample volume (1.5 l). The bromoform mixing ratio of one loop was calculated as the number of bromoform molecules of one loop as a fraction of number of molecules in a sample multiplied by $10^{12}$ to yield ppt."

To ensure that the reviewer can be fully confident in the calibration, a full copy of the calibration data and calculation procedure can be found in the supplementary section.

4) *Diurnal variation..p13. The authors suggest that increased photolysis of bromoform during the day is a major factor in the observed diurnal variability. However, I think that photolysis would be only a minor factor given the approximately 3 week lifetime of bromoform, which suggests that perhaps 5%/day is lost by photolysis. Thus even a constant source of bromoform would show only a minor diurnal cycle due to photolysis. Variation in source location, emission, and mixing offer much more reasonable hypotheses for the observed changes. Unfortunately, the authors seem to want to test every known factor that can influence bromoform concentrations, but they don't have sufficient data to adequately rule out or confirm any particular factor. This is, unfortunately, the weakness of the paper. Still, the data are useful and the discussion, while it could be improved, is not unreasonable.*

**Response:** We would like to thank the reviewer for pointing this out. We agree that the photolysis argument used here is overstated.

**Remedial action:** The diurnal section has been reworked as per the suggestions of both reviewers to indicate that variations in the MBL height are more locally significant on the time scale of the sampling and therefore the likely driver of the changes observed in the bromoform mixing ratios.

5) *Anthropogenic sources. It is clear from the Rn and CO measurements that the Cape Point site samples air with continental and anthropogenic influence. I think the authors still claim that anthropogenic sources can enhance the bromoform mixing ratios seen at Cape Point. I don't believe that the authors make any credible case for this influence, and perhaps the entire sections on "events" might be removed since no clear conclusions can be found to explain the variation in the data. For example, the event on the 18th of October is suggested to be influenced by emissions from the nuclear power plant cooling waters. This is due to the trajectory analysis and the observation of the highest CHBr3 level seen. However, as I look at the data, I note that: 1) The maximum on the 18th is not so different from that on the 17th or during other periods of marine biogenic influence only. 2) The trajectories on the 17th are very different from some on the 18th, with concentrations very comparable. 3) The authors suggestion implies that emissions from the Koeberg reactor would produce atmospheric CHBr3 levels much higher than the high peaks observed from marine only influence. It would be helpful to have some reasonable calculations, at least, to show that this might be possible.4) Based on the time series, it seems that the Rn/CO peak actually precedes the CHBr3 peak.*

**Response:** What we tried to do in response to the previous criticisms was separate the data into two groups: (i) those where anthropogenic influence could definitely be ruled out and (ii) those in which it could not be conclusively ruled out. Because a point falls into the 2$^{nd}$ group does not imply we have proved an anthropogenic contribution, but merely that we can conclusively exclude the possibility. Given the highly limited datasets when analysed on an event scale or case study we possibly over-interpreted the data.

**Remedial action:** As per the reviewer's suggestion, the case studies section as a whole has been removed from the text. This section was creating more confusion than shedding light on the measurements in this data sparse region and no clear conclusions can be found to explain the variations observed.

6) *Additional comments…1) While kelp are known sources of bromoform, there may certainly be other sources near coastlines. I say this from observations near my institution which contain very high levels of dissolved bromoform and high atmospheric levels (around 15 to 25 pptv) with no evidence of kelp in the area. We haven't yet investigated the specific sources yet. 2) If the authors continue with this work, it would be helpful to their analyses of different sources if they could measure a wider range of trace gases from their samples. The technology to do this sort of analysis is well established.*

**Response:** The authors would like to thank the reviewer for this comment. We agree with the points raced by the reviewer. From personal conversations with Dickon Young and Simon O'Doherty of the University of Bristol it is understood the marshes surrounding Mace Head

contribute significantly to the local atmospheric loading of bromoform. This is clearly not from an oceanographic source. While there are no significant marshes surrounding Cape Point, it is possible that the local vegetation may contribute to the atmospheric loading. There is extensive, untested, vegetation surrounding Cape Point to the north of the station. The natural reserve extends for at least 15 km N of the station.

**Remedial action:** Further work in this region is being planned and will make use of updated equipment and include a number of related compounds, both biogenic and anthropogenic. Hence, the long gap between this data and a new campaign being initiated.

**REVIEWER 2**

*The manuscript is much improved and I commend the authors on their hard work and perseverance. This still represents a rather limited dataset, but the authors have, in my opinion, made sensible alterations to their interpretation of the data and to the conclusions drawn. I still have some reservations, particular in the interpretation of some of the case studies, but I would be happy to recommend publication in ACP once the issues outlined below have been addressed.*

**Response:** The authors would like to thank the reviewer for their comments and suggestions regarding our paper and for the compliment regarding our hard work! Their comments have resulted in a significant improvement of the quality of the text and content of this manuscript.

> *P1, L5: replace "an" with "a"*

**Remedial action:** This has been corrected.

> *P1, L12: apart from a few minor anthropogenic sources ....*

**Remedial action:** The sentence has been revised to remove the anthropogenic sources clause. This has been formed into a separate sentence.

> *P1, L12: are there individual references for the different anthropogenic sources or do they all come from Quack and Wallace (2003)?*

**Response:** The reviewer is correct, there are a number of papers relating to anthropogenic sources of bromoform.

**Remedial action:** Selected authors describing different anthropogenic sources have been added to the text.

> *P2, L30: higher atmospheric levels are not just seen in "tropical" regions – they are often associated with seaweeds at mid-latitudes as well (Mace Head, etc).*

**Response:** Agreed, it is not only in the tropics that higher atmospheric levels of bromoform are found.

**Remedial action:** The section has been reworked to remove the ambiguity or the focus on the tropics. That elevated levels of bromoform can be produced in kelp beds, wherever they are and not just in the tropics.

> *P3, L20: replace full stop with a comma – i.e. ".. into the UTLS, where bromine-*

*initiated ..."*

**Remedial action:** This has been done.

> *P3, L21: add "the". i.e. " ... contribution from the Cape Point region is ...."*.

**Remedial action:** This has been corrected as the reviewer suggested.

> *P3, L31: delete second use of the word "extensive"*

**Remedial action:** This has been removed.

> *P4, L11: typo "Perkin Elmer", not "Perkin Elmar".*

**Remedial action:** The typo has been corrected.

> *P8, L13: what does pm mean (13.2 pm 9.7 ppt)? plus/minus?*

**Response:** This was a typographical error in Latex. The command instruction was left out. The reviewer is correct it is meant to be plus/minus.

**Remedial action:** This has been corrected.

> *P8, L13: Figure number is missing*

**Remedial action:** This has been corrected.

> *P8, L14: I don't particularly like this sentence "The measurements were largely consistent within a few days, however could vary by 10s of ppt between days". What are the authors trying to say here? Please try rephrasing. Something like "Bromoform was typically in the range of 1-20 ppt but on several occasions elevated mixing ratios were encountered that could last for several hours ....."*

**Response:** The authors would like to thank the reviewer for this comment. Their suggestion is excellent and has been implemented as suggested.

**Remedial action:** The sentence has been revised as per the reviewer's suggestion.

> *P8, L26 Should "Cape Town" read "Cape Point"?*

**Response:** The reviewer is correct, this should read "Cape Point"

**Remedial action:** This sentence has been revised and corrected.

> *P8, L28: missing parentheses around the reference.*

**Remedial action:** This has been corrected.

> *P9, Fig 3: Units are missing from the y-axis*

**Response:** The authors would like to thank the reviewer for this comment.

**Remedial action:** Where possible the units have been added directly to the figure, otherwise

they have been included in the caption text.

*P9, Fig.3: please make it clearer either in the Figure caption or perhaps in the text which is high tide and which is low tide. The graph ($2^{nd}$ panel down) varies from 0.6 to 1.4, but what does this mean?*

**Response:** We apologise for the confusion caused in this figure. The tide heights are given in metres above a chart datum defined by the South African Hydrographic Office. Thus, lower values denote a low tide while higher values indicate a high tide.

**Remedial action:** The caption of figure with tides has been amended to reflect this. Furthermore, a more complete description of the source of the tidal data and what it means has been added to the Methods and Material section.

*P9, L4-6: You say that the tidal height is a "necessary but not sufficient factor" in the high bromoform episodes but then go on to say that "it is therefore likely that the extensive local kelp beds are an important source of the observed bromoform". These statements seem to contradict themselves a little. Please consider rephrasing these 2 sentences.*

*It is interesting that the seaweeds do not become completely exposed at Cape Point. Is this true of the wider region as well? Are you able to smell the seaweeds at low tide? This might be a good indicator of very local emissions!*

**Response:** We thank the reviewer for observing this ambiguity in the text. The sentences have be revised to as per their suggestion to remove ambiguity and contradiction. This will greatly improve the quality of the text.

It is true of the region as a whole. The horizontal extent of the tides in South Africa is quite limited. One can definitely smell the seaweeds at low tide, especially an iodine smell. This is particularly noticeable at low wind speeds.

**Remedial action:** The sentences have been revised and now read:

"While the maximum tidal range in the vertical at Cape Point is comparable to that at Mace Head, the horizontal extent is much smaller, may explain the lack of correlation. Consequently, during low tide at Cape Point, only the tops of the kelp fronds become exposed to the atmosphere. This is common around the coast of South Africa. Nonetheless, the elevated bromoform events with the highest mixing ratios all appear to mostly occur shortly after low tide (Fig. 3). It is therefore likely that the extensive local kelp beds are an important source of the bromoform observed at the station."

*P10, L3: replace "particularly" with "including"*

**Remedial action:** This correction has been made.

*P11, L1: missing Table number*

**Remedial action:** The missing table number has been inserted.

*P11, L2-4: I don't like this sentence very much either. "The introduction of ........". How does it allow for the determination of scale of the anthropogenic contributions in this*

*region? Do you mean anthropogenic bromoform? As I mentioned in my first review when you have seaweed beds to the north of Cape Town it is very hard to distinguish whether the CHBr3 comes from an anthropogenic source rather than a marine source further to the north.*

**Response:** The statement was meant to be more general about air masses rather than about bromoform in particular. We apologise for the confusion caused. The reviewer is correct that the kelp beds to the north make the attribution of sources difficult. The local measurement of radon at Cape Point has been used extensively as a marker of air mass characterisation. An air mass sample from the north could entrain bromoform from a biogenic source on its way to Cape Point. An analysis of the ratio of $CH_2Br_2$ to $CHBr_3$ could shed light on the source of the air mass, or whether there has been anthropogenic entrainment.

**Remedial action:** The sentence regarding the introduction of intermediate air has been revised to make it clear that it is anthropogenically modified in general. We feel that the description on air mass characterisation is sufficiently clear that we are able to separate clean marine air from anthropogenically modified air at Cape Point. Furthermore, that in this case we are not able to separate or identify whether there has been any anthropogenic modification to the bromoform measured at Cape Point.

> *P11, L9: strictly speaking the kelp beds were not "responsible" for the measurements. Please rephrase.*

**Response:** The reviewer is correct that the kelp beds were not 'responsible' for the measurements.

**Remedial action:** The sentence has been revised and now reads:

"It is therefore, likely that the local kelp beds were the source of the bulk of the measured bromoform, including the elevated mixing ratios observed."

> *P11, L24-25: although you say there was no correlation between bromoform and boundary layer height you cannot say that BL ht has no influence. Even in the diurnal cycle shown in Figure 4, the gradually declining concentrations after 10 am could partly be due to an expanding boundary layer as the atmosphere warms up. This would cause a dilution and therefore contribute to the decline. Similarly in the evening when the nocturnal boundary forms, might this not contribute to the higher concentrations you observe in the early evening and through to 11 pm. There could also be a link between boundary layer height and tide. Low tide and low BL could lead to higher concentrations particularly if there are emissions at night?*

**Response:** We thank the reviewer for these comments and suggestions.

**Remedial action:** The meteorological and diurnal sections have been revised to include a more complete description of the role of the MBL in the observed bromoform variations.

> P12, L5: delete "in"

**Remedial action:** This has been deleted.

> P12, L6: the overnight low looks more like 12-13 ppt from Figure 4

**Response:** The author is correct, that the wrong value had been reported.

**Remedial action:** This has been corrected with the revised values based on the updated calibration.

> P12, Fig 4: it would be useful to have an indication as to the number of samples in each hourly bin

**Response:** The reviewer is thanked for this comment.

**Remedial action:** The following table has been calculated, as per the reviewers suggestions. However, perhaps this represents more detail than is necessary in the final publication. We are happy to include/no include as per the editor's discretion.

**Table:** Calculated bromoform mixing ratios and 95% confidence intervals for sampling period to create mean diurnal pattern. Measurements of bromoform are given in ppt.

| Hour of Day | Mean | Upper | Lower | Number of samples |
|---|---|---|---|---|
| 6 | 21.0 | 27.6 | 16.3 | 3 |
| 7 | 17.9 | 25.1 | 11.2 | 8 |
| 8 | 18.5 | 22.1 | 14.8 | 8 |
| 9 | 20.0 | 24.3 | 13.7 | 8 |
| 10 | 24.2 | 39.9 | 12.5 | 5 |
| 11 | 29.1 | 44.4 | 12.2 | 5 |
| 12 | 20.6 | 28.8 | 12.9 | 11 |
| 13 | 26.7 | 38.1 | 17.3 | 12 |
| 14 | 20.6 | 31.0 | 13.3 | 10 |
| 15 | 25.8 | 36.1 | 17.2 | 9 |
| 16 | 18.1 | 28.0 | 11.0 | 9 |
| 17 | 12.7 | 18.6 | 6.7 | 8 |
| 18 | 19.4 | 27.5 | 10.9 | 6 |
| 19 | 17.8 | 26.6 | 8.8 | 9 |
| 20 | 19.3 | 26.0 | 11.6 | 7 |
| 21 | 19.1 | 26.5 | 11.7 | 8 |
| 22 | 16.8 | 26.6 | 5.7 | 7 |
| 23 | 26.8 | NA | NA | 2 |

> *P13, para 1: note my point about BL height (above).*

**Response:** The reviewer is thanked for their comments regarding the MBL.

**Remedial action:** The meteorological and diurnal sections have been revised to include a better description of the variability of the MBL and the effect on bromoform concentrations.

> *P15, Fig 6: Does the sample with the maximum concentration of bromoform correspond to the back- trajectory that passes through Koeberg?*

**Response:** In short, no. It does pass, at low level, over an extensive water treatment facility though. Also, interesting to note is that later trajectories do pass over Koeberg.

**Remedial action:** These are calculated trajectories for 18 October 2011 for every three hours. This section has been removed from the manuscript. The elevated mixing ratio reported in Fig 6 occurred at approx. 11 am. One can see that the trajectories arriving between 9 am and 12 pm occur from an easterly direction, while later trajectories come from the north and transit over Koeberg.

**Figure:** Calculated back trajectories starting every three hours on 18 October 2011.

[Figure]

*P15, Fig 6: why do the trajectories start at an altitude of over 200m? Is this the altitude of the sampling location (I had assumed it was lower)? Also, the trajectories look a bit odd as they seem to go to almost negative altitudes, particularly on the 18th? You do not comment on these altitude profiles in the text. What do they actually tell us?*

**Response:** The reviewer is correct, the South African Weather Service GAW station sits at the top of a cliff at a height of 230 m. The heights on the 18th are close to zero, but not actually negative. The size and shape of the markers might make it look like the heights transition below the zero line.

The trajectory heights, and maybe this was not made clear in the text, help to show whether surface sources might be entrained in the air mass. For example, had the trajectory height on 18 October been at 500 or 1000 m then a source at Koeberg could not have been entrained.

**Remedial action:** The height of the laboratory has been added into the introductory text. The case studies section has been removed entirely from the manuscript. This includes this back trajectory analysis. A revised analysis involving a concentration weighted trajectory (CWT) model analysis has been included.

> P16, Fig 7: the line for mbl looks a little odd. Was there no change in boundary layer height on the 26 October? Also, the units are missing from the y-axis (same for Figs 5 and 8).

**Response:** We thank the reviewer for this comment. There were occasions when the calculated MBL height did not vary between balloon runs. This has been revised.

**Remedial action:** The MBL heights have been checked and revised. Updated figures have been plotted for this revision of the manuscript.

> P17, L7: "The" should read "the" (no capital required).

**Response:** We thank the reviewer for this observation.

**Remedial action:** The word 'the' has been corrected to remove the unnecessary capital.

> P17, L25: missing word "between 3 and 6 pm"?

**Remedial action:** The missing word 'between' has been added.

> P18, L5: delete the first "known", and, better still, replace "known" with "potential". You haven't confirmed in this work that the power and water plants in CT actually produce bromoform.

**Response:** The authors thank the reviewer for this comment. Although we have not proved that nuclear power plants produce significant levels of bromoform other authors have, e.g. Quack and Suess, 1999. It is therefore reasonable to assume that Koeberg is a source, at least locally, of atmospheric bromoform.

**Remedial action:** The case studies have been removed as per the reviewer's suggestion. Thus, negating the need for corrections on this point.

> P18, Fig 8: again the boundary layer height looks strange. No change between 9 am on the 7$^{th}$ Nov and 9 am on 8$^{th}$ Nov?

**Remedial action:** The case studies and figures have been removed from the text.

> P19, Fig 9: The wrong date is used in the figure caption.

**Remedial action:** The figure and caption have been removed.

**Comment on the case studies**

*I understand that the authors are trying to highlight some of the more interesting features in their data, but I worry that they do not really have enough data to come to any conclusions. For example,*

*in Case 1, the argument for an anthropogenic source is essentially based on one single data point, which occurs during a period of elevated CO and radon (and also at low tide). I wonder if the trajectories in Figure 6 could be coloured differently to show the gradual change of air mass origin over the period.*

**Response:** See remedial action below.

*In Case 2 it is very hard to discern anything meaningful from the various parameters discussed, particularly in regard to the tidal heights. I do however notice that the wind speed increases over the period. As the winds are coming from the west, would an open ocean source (influenced by increasing winds) not be a possibility as well? Again this would be highly speculative.*

**Response:** The reviewer is correct that the data in this study is sparse. This makes it difficult to draw any firm conclusions. The limited data leaves little scope but to be speculative.

**Remedial action:** A decision has been made to remove the case studies section from the manuscript, as per both reviewers' suggestion. The paper focuses on the measurements of bromoform at Cape Point as a whole, the diurnal cycle and the likely impacts of variations in the MBL height and a back trajectory analysis which suggests that the highest concentrations might be sourced from off shore in the south Atlantic.

[revised manuscript text omitted]

---

## Author Response (AR3)

The authors would like to thank the reviewer for their time and effort in reviewing both the manuscript and the data and calibration method. The comments and suggestions here have greatly improved the quality of this paper.

> *The manuscript by Kuyper et al. is greatly improved in its organization and presentation of the results. However, I have found some issues that I think need additional clarification. These are related to the analytical methods and calibration. I thank the authors for providing an actual spreadsheet file of their standards and air analysis. This definitely helped me understand the procedure much better, but also brought up some new issues. In my view, though the authors show a good fit to the bromoform peak area vs mixing ratio calibration, the random errors are very large. Furthermore, there are a few statements in the description that are not accurate.*

**Response:** The reviewer raises some fantastic suggestions and points. These will greatly improve the quality and scientific merit of the manuscript. We are glad that the data was able to elucidate matters regarding paper. The random errors in these measurements are quite large. This is likely due to the manual nature of the system, trapping to injection and oven temperature profile adjustments. Although these are large we feel that the data are still interesting and merits publication as a first approximation of the range of values found in this region, even if they should be treated with a degree of caution.

> *First, a linear regression is not a measure of accuracy, so the claim of 99.6% accuracy is not appropriate. In terms of regression analysis, I put the authors' data, not including the measurements from August, into a statistical program (SYSTAT) and found an R^2 of 0.86 and a regression line (similar to the authors') of Area = 129 (-+168) + 227 (-+7.5) \* Std Mixing Ratio, where the -+ is the standard error of the estimate. The regression analysis was used by the authors to calculate all of the air measurements. The authors apparently did not use the standards interspersed with the samples to calculate any drift as claimed in the manuscript. The same area/mixing ratio relationship was used for all calculations. If the standard responses were used for drift calculation, then the calculated data would be much different.*

**Response:** The authors would like to apologise for any confusion caused in stating that the regression line is a measure of accuracy. We agree that linear regression is not a measure of accuracy. A correlation of mixing ratio and peak area is a measure of accuracy. The text has been updated to correct this error.

The data has been rerun in R using all the data we agree with the reviewer. The $R^2$ of the data provided is 0.86. We have further updated the calibration data as per the reviewer's comments and determined an overall $R^2$ 0.82, based on the correlation of peak area to mixing ratio. We apologise for the error and overstating the accuracy of the measurements. The original $R^2$ value had been calculated based on the mean value of each loop injection. This reduced the uncertainty. The correct overall values have been added to the text as well as updated.

We had used the single regression analysis to calculate the air mixing ratios. This has been corrected. A complete calibration was run at the start of the sampling campaign. Thereafter, a calibration point was run approximately daily. After the initial calibration the approximately daily calibration points were coerced into a regular matrix. An interpolated using a 3 point

running mean was then used to close the gaps between calibration points. The running mean incorporated new data points. This resulted in an approximately 8 hour window between calibration points. The slope and intercept values were updated for each 8 hour time step. The air measurements were then calculated from these updated regression values, at each time step. This has been done and added to the text. The revised calibration using the interpolated calculations does not seem to have resulted in any significant shifts within the air data as reported previously. The results and discussion have been updated to include revised figures and text. This is, as stated previously, largely unchanged. Given the limited nature of the system and the available data this seemed the most rigorous available method to address the gaps in the calibration.

> *One point regarding this method is that a zero area produces a mixing ratio of 0.83 pptv. It may be worth pointing that out.*

**Response:** The authors would like to thank the reviewer for this. We have noted this and accommodated that in the regression calculations.

> *There are a number of troubling aspects to the calibration that suggest problems with their technique. The variability of each standard level is remarkably high, with ranges over a factor of 2 or more in the response factor at each level. Another issue is the variability of the blank. As indicated on the spreadsheet about 25% of the blanks produced measurable peak areas, and these range from 1.36 to 8.85 pptv bromoform. Clearly there was some issue with carryover or some other problem that needs to be examined. The next major issue is that the response variability at the 21 ppt level (near the mean ambient concentration) is about 55% for one standard deviation. Since the authors have no data on replicate ambient analysis, one must assume that the variability of the ambient data is also in this range. An overall variation in the response factor considering all standards is 43% (1 SD), still remarkably high, and this variability must be considered for the ambient samples. Or, if the authors think the analytical system for ambient samples somehow has a better precision than that demonstrated by the loop injections of the standards, there should be some explanation of why. Or they should assume that the response variation in the standard injections reflects actual response variations of the detector and correct the ambient data for that response change. For the future, my suggestion would be to work with one or more secondary ambient air samples that are calibrated against a primary standard. (And to determine the source of the variability in the loop injections!).*

**Response**: With regards to the issues of carry over, this issue was dealt with by analysing 'blank' $N_2$ samples are regular intervals to check for carry over. These were done every day before sampling and after every calibration point. Carry over was seldom noticed following a calibration point, however, as the reviewer notes it did happen on occasion. When it was noticed a second blank was run to ensure no artificially $CHBr_3$ was introduced in the measurements. Moreover, carry over between ambient measurement and subsequent nitrogen blank analysis averaged 4.8 % and was typically zero. From this and from the regular daily analysis of nitrogen blanks we believe we can confidently infer that carry over

from calibration standard to follow ambient air sample was impossible and from ambient to next ambient was negligible for these data.

The variability of the standards is a concerning factor. This variability may stem from manual nature of the entire system. The system was very rudimentary, involving manual injection of standards on to the trap and injection into the system. As there are fewer manual steps involved in an air sample compared to a standard injection, it is possible that the samples might be more accurate than the standards. The trap was also comparatively large (5 mm OD), this might have resulted in differing adsorption and desorption rates, contributing to the variability. We agree with the reviewer that there is a large degree of uncertainty within this data and while the data should be treated with a fair degree of caution they do provide an informative first fixed point picture of the measurements in this region. Further work, on a more up to date system is currently being done to refine these values.

> *The authors note that the drift was measured from day to day using the 1 − 3 loop injections. However, this day to day "drift", sometimes calculated from one standard only and sometimes from multiple standards, varies over a factor of 3. Apparently, though this daily "drift" was not taken into account in the calculations. This should be discussed in the methods section.*

**Response**: The calculations for the drift have been updated in the data and the text. The text has been updated to reflect the time and spacing between standard samples. Please see above.

> *Another discrepancy, based on the spreadsheet information, is the frequency of the standard injections. The manuscript describes standard injections every five samples. However, this does not seem to be the case for each day. Some days have no standard injections, others multiple, and others with 7 − 8 samples run with no standard injection. Typically, no standard injection ended the daily runs. Please correct this description of standardization frequency in the text.*

**Response**: The text has been corrected to reflect that the gaps between calibration points were irregular, although usually daily. A method has been implemented to interpolate between calibration points. This creates a regular matrix against which the air measurements have been calculated. This incorporates the standard injections and adjusts for drift at each time step.

> *Finally, I am curious about data that appeared in the spreadsheet, but was not included in the manuscript. The spreadsheet shows several values of bromoform > 60 ppt that are not shown in the manuscript, and apparently were also not included in calculating sample average. It is not uncommon to have "flyers" sometimes in the data, but it would be helpful to mention if some data are excluded.*

Response: The reviewer is correct, there were outliers that were removed from data shown in the text. We apologise that this was omitted from the text. An interquartile range (IQR) method was used to identify and then remove outliers. Outliers were defined as values greater or less than $Q_{75/25} \pm 1.5$ x IQR (Underhill and Bradfield, 2005). This worked out to be

71.4 ppt and -20.8 ppt for the upper and lower bounds. The lower bound determined here is lower than the limit of detection. Therefore, the limit of detection was used as the lower limit for observations here. The mean was then re-calculated to a value of 24.8 ppt with a standard deviation of 14.8 ppt. The text has been amended to reflect that outlying values were removed or excluded and how these were determined.

Underhill, L. G. and Bradfield, D. (2005). *Introstat*. Juta Academic, Cape Town, South Africa.

> *I thank the authors again for sharing their raw data with the reviewers. Based on this data, I request a more thorough and accurate description of the analytical procedures, especially related to calibration and calculation, and how the variability observed impacts the uncertainties associated with the reported measurements. At this stage, I can't be sure that the lack of drift correction hasn't masked features in the data, and I am also unsure about the absolute calibration, which makes it difficult to compare this data with others. If these measurements are to be published, there needs to a much clearer description of the various uncertainties and limitations.*

**Response**: The authors would like to thank the reviewer for their comments and suggestions. The text has been amended to greater reflex the uncertainty of the measurements. The data now reflects drift and should not be masking anything in the data.

There is unfortunately no way to expand the calibration to include an non-absolute parameter. We agree with the reviewer that this does make the comparison of this data with others measurements challenging. Although the uncertainty is large, these are the first fixed point data at Cape Point. The values proved a first approximation of the values in this region. Further and updated measurements will improve the accuracy of these measurements.

[revised manuscript text omitted]

---

## Author Response (AR4)

Dear Dr. Hofzumahaus,

Please find below a detailed description of the change made to the manuscript as requested by the reviewer. The authors would like to thank the reviewer for their time and effort in reviewing the changes and responses to our manuscript. The reviewer's suggestion will greatly improve the readability and understanding of the science.

Reviewer's comment
*I appreciate the authors' efforts to clarify the measurement technique and report errors. However, I think the authors should clearly acknowledge the limitation of the method that they used with similar text as they provided in their response:*

*"The random errors in these measurements are quite large. This is likely due to the manual nature of the system, trapping to injection and oven temperature profile adjustments. Although these are large we feel that the data are still interesting and merits publication as a first approximation of the range of values found in this region, even if they should be treated with a degree of caution. "*

*This will make clear that the current report provides a reasonable approximation of bromoform in this interesting area, but improvements in analytical performance is needed. I know that it's tough as an author to actually write these limitations, but necessary in this case, I think.*

**Response:** The authors would like to thank the reviewer for their suggestion. We agree with the reviewer that a statement in the text to the extent of the uncertainty of the measurements is warranted. Text has been added to the Conclusions describing  the uncertainty associated with the measurements.
The start of the conclusions now reads:

**Conclusions**
The data presented here represents the first fixed point quantitative atmospheric bromoform measurements at the Cape Point Global Atmospheric Watch Station, but also the first such dataset in southern Africa. The 135 discrete measurements made over the course of October/November 2011 exhibited a mean bromoform mixing ratio of $24.8 \pm 14.8$ ppt. The maximum bromoform mixing ratio reported here (64.6 ppt) was consistent with past studies, for example: that reported in Cape Verde (43.7 ppt, OBrien2009) or New Hampshire (47.4 ppt, Zhou2008). **However, it should be noted that the random errors in these measurements are quite large, with a precision of 22.2%. The scale of these uncertainties is due to the manual nature of the system, trapping to injection and oven temperature profile adjustments. Although the uncertainty associated with the data presented here is large, we feel that the data are still interesting as a first approximation of the range of values found in this region. Given the uncertainty the data should be treated with a degree of caution.**

The authors would like to thank the reviewers who have given their time and effort to review the manuscripts and for their invaluable feedback.

[revised manuscript text omitted]